# DEQGAN: Learning the Loss Function for PINNs with Generative Adversarial Networks

## Abstract

Solutions to differential equations are of significant scientific and engineering rele-
vance. Physics-Informed Neural Networks (PINNs) have emerged as a promising
method for solving differential equations, but they lack a theoretical justification
for the use of any particular loss function. This work presents Differential Equation
GAN (DEQGAN), a novel method for solving differential equations using gener-
ative adversarial networks to "learn the loss function" for optimizing the neural
network. Presenting results on a suite of twelve ordinary and partial differential
equations, including the nonlinear Burgers', Allen-Cahn, Hamilton, and modified
Einstein's gravity equations, we show that DEQGAN[1] can obtain multiple orders
of magnitude lower mean squared errors than PINNs that use $L_2$, $L_1$, and Huber
loss functions. We also show that DEQGAN achieves solution accuracies that are
competitive with popular numerical methods. Finally, we present two methods to
improve the robustness of DEQGAN to different hyperparameter settings.

## 1 Introduction

In fields such as physics, chemistry, biology, engineering, and economics, differential equations are
used to model important and complex phenomena. While numerical methods for solving differential
equations perform well and the theory for their stability and convergence is well established, the
recent success of deep learning [3, 10, 17, 29, 40, 47, 52, 53] has inspired researchers to apply
neural networks to solving differential equations, which has given rise to the growing field of
Physics-Informed Neural Networks (PINNs) [19, 20, 35, 36, 42–44, 48, 50].

In contrast to traditional numerical methods, PINNs: provide solutions that are closed-form [30],
suffer less from the "curse of dimensionality" [16, 20, 43, 48], provide a more accurate interpolation
scheme [30], and can leverage transfer learning for fast discovery of new solutions [11, 13]. Further,
PINNs do not require an underlying grid and offer a meshless approach to solving differential
equations. This makes it possible to use trained neural networks, which typically have small memory
footprints, to generate solutions over arbitrary grids in a single forward pass.

PINNs have been successfully applied to a wide range of differential equations, but lack a theoretical
justification for the use of a particular loss function from the standpoint of predictive performance. In
domains outside of differential equations, data following a known noise model (e.g. Gaussian) have
clear justification for fitting models with specific loss functions (e.g. $L_2$). In the case of deterministic
differential equations, however, there is no noise model and we lack an equivalent justification.

To address this gap in the theory, we propose generative adversarial networks (GANs) [14] for solving
differential equations in a fully unsupervised manner. Recently, multiple works have shown that
adaptively modifying the PINN loss function throughout training can lead to improved solution

---

[1]We provide our PyTorch code at [link hidden to preserve anonymity]

Submitted to 36th Conference on Neural Information Processing Systems (NeurIPS 2022). Do not distribute.

accuracies [37, 57]. The discriminator network of our GAN-based method, however, can be thought of as "learning the loss function" for optimizing the generator, thereby eliminating the need for an explicit loss function and providing even greater flexibility than an adaptive loss. Beyond the context of differential equations, it has also been shown that where classical loss functions struggle to capture complex spatio-temporal dependencies, GANs may be an effective alternative [32, 26, 31].

Our contributions in this work are summarized as follows:

- We present Differential Equation GAN (DEQGAN), a novel method for solving differential equations in a fully unsupervised manner using generative adversarial networks.

- We highlight the advantage of "learning the loss function" with a GAN rather than using a pre-specified loss function by showing that PINNs trained using $L_2, L_1$, and Huber losses have variable performance and fail to solve the modified Einstein's gravity equations [7].

- We present results on a suite of twelve ordinary differential equations (ODEs) and partial differential equations (PDEs), including highly nonlinear problems, showing that our method produces solutions with multiple orders of magnitude lower mean squared errors than PINNs that use $L_2, L_1$, and Huber loss functions.

- We show that DEQGAN achieves solution accuracies that are competitive with popular numerical methods, including the fourth-order Runge-Kutta and second-order finite difference methods.

- We present two techniques to improve the training stability of DEQGAN that are applicable to other GAN-based methods and PINN approaches to solving differential equations.

## 2   Related Work

A variety of neural network methods have been developed for solving differential equations. Some of these are supervised and learn the dynamics of real-world systems from data [4, 9, 15, 44]. Others are semi-supervised, learning general solutions to a differential equation and extracting a best fit solution based on observational data [41]. Our work falls under the category of *unsupervised* neural network methods, which are trained in a data-free manner that depends solely on the equation residuals. Unsupervised neural networks have been applied to a wide range of ODEs [13, 30, 34, 36] and PDEs [20, 43, 48, 50], primarily use feed-forward architectures, and require the specification of a particular loss function computed over the equation residuals.

Goodfellow et al. [14] introduced the idea of learning generative models with neural networks and an adversarial training algorithm, called generative adversarial networks (GANs). To solve issues of GAN training instability, Arjovsky et al. [2] introduced a formulation of GANs based on the Wasserstein distance, and Gulrajani et al. [18] added a gradient penalty to approximately enforce a Lipschitz constraint on the discriminator. Miyato et al. [39] introduced an alternative method for enforcing the Lipschitz constraint with a spectral normalization technique that outperforms the former method on some problems.

Further work has applied GANs to differential equations with solution data used for supervision. Yang et al. [56] apply GANs to stochastic differential equations by using "snapshots" of ground-truth data for semi-supervised training. A project by students at Stanford [51] employed GANs to perform "turbulence enrichment" of solution data in a manner akin to that of super-resolution for images proposed by Ledig et al. [32]. Our work distinguishes itself from other GAN-based approaches for solving differential equations by being *fully unsupervised*, and removing the dependence on using supervised training data (i.e. solutions of the equation).

## 3   Background

### 3.1   Unsupervised Neural Networks for Differential Equations

Early work by Dissanayake & Phan-Thien [12] proposed solving initial value problems in an unsupervised manner with neural networks. In this work, we extend their approach to handle spatial domains and multidimensional problems. In particular, we consider general differential equations of the form

$$F\left(t, \mathbf{x}, \Psi(t, \mathbf{x}), \frac{d\Psi}{dt}, \frac{d^2\Psi}{dt^2}, \ldots, \Delta\Psi, \Delta^2\Psi, \ldots\right) = 0 \tag{1}$$

where $\Psi(t, \mathbf{x})$ is the desired solution, $d\Psi/dt$ and $d^2\Psi/dt^2$ represent the first and second time derivatives, $\Delta\Psi$ and $\Delta^2\Psi$ are the first and second spatial derivatives, and the system is subject to certain initial and boundary conditions. The learning problem can then be formulated as minimizing the sum of squared residuals (i.e., the squared $L_2$ loss) of the above equation

$$\min_\theta \sum_{(t,\mathbf{x}) \in \mathcal{D}} F\left(t, \mathbf{x}, \Psi_\theta(t, \mathbf{x}), \frac{d\Psi_\theta}{dt}, \frac{d^2\Psi_\theta}{dt^2}, \ldots, \Delta\Psi_\theta, \Delta^2\Psi_\theta, \ldots\right)^2 \tag{2}$$

where $\Psi_\theta$ is a neural network parameterized by $\theta$, $\mathcal{D}$ is the domain of the problem, and derivatives are computed with automatic differentiation. This allows backpropagation [22] to be used to train the neural network to satisfy the differential equation. We apply this formalism to both initial and boundary value problems, including multidimensional problems, as detailed in Appendix A.2.

## 3.2 Generative Adversarial Networks

Generative adversarial networks (GANs) [14] are generative models that use two neural networks to induce a generative distribution $p(x)$ of the data by formulating the inference problem as a two-player, zero-sum game.

The generative model first samples a latent random variable $z \sim \mathcal{N}(0, 1)$, which is used as input into the generator $G$ (e.g., a neural network). A discriminator $D$ is trained to classify whether its input was sampled from the generator (i.e. "fake") or from a reference data set (i.e. "real").

Informally, the process of training GANs proceeds by optimizing a minimax objective over the generator and discriminator such that the generator attempts to trick the discriminator to classify "fake" samples as "real". Formally, one optimizes

$$\min_G \max_D V(D, G) = \min_G \max_D \mathbb{E}_{x \sim p_{\text{data}}(x)}\left[\log D(x)\right] + \mathbb{E}_{z \sim p_z(z)}[1 - \log D(G(z))] \tag{3}$$

where $x \sim p_{\text{data}}(x)$ denotes samples from the empirical data distribution, and $p_z \sim \mathcal{N}(0, 1)$ samples in latent space [14]. In practice, the optimization alternates between gradient ascent and descent steps for $D$ and $G$, respectively. Further details on training and architecture are provided in Appendix A.4.

## 3.3 Guaranteeing Initial & Boundary Conditions

Lagaris et al. [30] showed that it is possible to exactly satisfy initial and boundary conditions by adjusting the output of the neural network. For example, consider adjusting the neural network output $\Psi_\theta(t, \mathbf{x})$ to satisfy the initial condition $\Psi_\theta(t, \mathbf{x})\big|_{t=t_0} = x_0$. We can apply the re-parameterization

$$\tilde{\Psi}_\theta(t, \mathbf{x}) = x_0 + t\Psi_\theta(t, \mathbf{x}) \tag{4}$$

which exactly satisfies the initial condition. Mattheakis et al. [36] proposed an augmented re-parameterization

$$\tilde{\Psi}_\theta(t, \mathbf{x}) = \Phi\left(\Psi_\theta(t, \mathbf{x})\right) = x_0 + \left(1 - e^{-(t-t_0)}\right)\Psi_\theta(t, \mathbf{x}) \tag{5}$$

that further improved training convergence. Intuitively, Equation 5 adjusts the output of the neural network $\Psi_\theta(t, \mathbf{x})$ to be exactly $x_0$ when $t = t_0$, and decays this constraint exponentially in $t$. Chen et al. [8] provide re-parameterizations to satisfy a range of other conditions, including Dirichlet and Neumann boundary conditions, which we employ in our experiments and detail in Appendix A.2.

## 4 Differential Equation GAN

In this section, we present our method, Differential Equation GAN (DEQGAN), which trains a GAN to solve differential equations in a *fully unsupervised* manner. To do this, we rearrange the differential equation so that the left-hand side ($LHS$) contains all the terms which depend on the generator (e.g. $\Psi$, $d\Psi/dt$, $\Delta\Psi$, etc.) and the right-hand side ($RHS$) contains only constants (e.g. zero).

During training, we sample points from the domain $(t, \mathbf{x}) \sim \mathcal{D}$ and use them as input to a generator $G(x)$, which produces candidate solutions $\Psi_\theta$. We sample points from a noisy grid that spans $\mathcal{D}$,

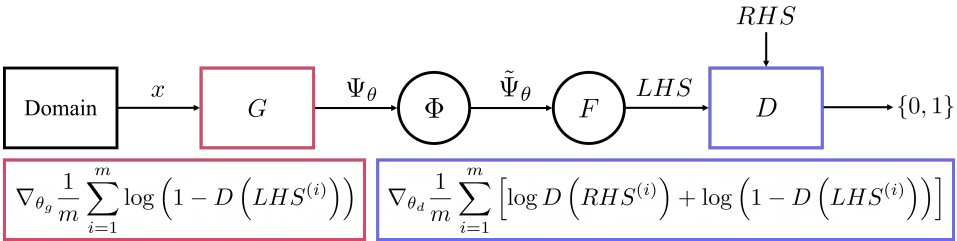

Figure 1: Schematic representation of DEQGAN. We pass input points $x$ to a generator $G$, which produces candidate solutions $\Psi_\theta$. Then we analytically adjust these solutions according to $\Phi$ and apply automatic differentiation to construct $LHS$ from the differential equation $F$. $RHS$ and $LHS$ are passed to a discriminator $D$, which is trained to classify them as "real" and "fake," respectively.

which we found reduced interpolation error in comparison to sampling points from a fixed grid. We then adjust $\Psi_\theta$ for initial or boundary conditions to obtain the re-parameterized output $\tilde{\Psi}_\theta$, construct the $LHS$ from the differential equation $F$ using automatic differentiation

$$LHS = F\left(t, \mathbf{x}, \tilde{\Psi}_\theta(t, \mathbf{x}), \frac{d\tilde{\Psi}_\theta}{dt}, \frac{d^2\tilde{\Psi}_\theta}{dt^2}, \dots, \Delta\tilde{\Psi}_\theta, \Delta^2\tilde{\Psi}_\theta, \dots\right) \tag{6}$$

and set $RHS$ to its appropriate value (in our examples, $RHS = 0$). Training proceeds in a manner similar to traditional GANs. We update the weights of the generator $G$ and the discriminator $D$ according to the gradients

$$g_G = \nabla_{\theta_g} \frac{1}{m} \sum_{i=1}^{m} \log\left(1 - D\left(LHS^{(i)}\right)\right), \tag{7}$$

$$g_D = \nabla_{\theta_d} \frac{1}{m} \sum_{i=1}^{m} \left[\log D\left(RHS^{(i)}\right) + \log\left(1 - D\left(LHS^{(i)}\right)\right)\right] \tag{8}$$

where $LHS^{(i)}$ is the output of $G\left(x^{(i)}\right)$ after adjusting for initial or boundary conditions and constructing the $LHS$ from $F$. Note that we perform stochastic gradient *descent* for $G$ (gradient steps $\propto -g_G$), and stochastic gradient *ascent* for $D$ (gradient steps $\propto g_D$). We provide a schematic representation of DEQGAN in Figure 1 and detail the training steps in Algorithm 1.

---

**Algorithm 1** DEQGAN

**Input:** Differential equation $F$, generator $G(\cdot; \theta_g)$, discriminator $D(\cdot; \theta_d)$, grid $x$ of $m$ points with spacing $\Delta x$, perturbation precision $\tau$, re-parameterization function $\Phi$, total steps $N$, learning rates $\eta_G, \eta_D$, Adam optimizer [27] parameters $\beta_{G1}, \beta_{G2}, \beta_{D1}, \beta_{D2}$
**for** $i = 1$ **to** $N$ **do**
    **for** $j = 1$ **to** $m$ **do**
        Perturb $j$-th point in mesh $x_s^{(j)} = x^{(j)} + \epsilon, \epsilon \sim \mathcal{N}(0, \frac{\Delta x}{\tau})$
        Forward pass $\Psi_\theta = G(x_s^{(j)})$
        Analytic re-parameterization $\tilde{\Psi}_\theta = \Phi(\Psi_\theta)$
        Compute $LHS^{(j)} = F\left(t, \mathbf{x}, \tilde{\Psi}_\theta(t, \mathbf{x}), \frac{d\tilde{\Psi}_\theta}{dt}, \frac{d^2\tilde{\Psi}_\theta}{dt^2}, \dots, \Delta\tilde{\Psi}_\theta, \Delta^2\tilde{\Psi}_\theta, \dots\right)$
        Set $RHS^{(j)} = 0$
    **end for**
    Compute gradients $g_G, g_D$ (Equation 7 & 8)
    Update generator $\theta_g \leftarrow \texttt{Adam}(\theta_g, -g_G, \eta_G, \beta_{G1}, \beta_{G2})$
    Update discriminator $\theta_d \leftarrow \texttt{Adam}(\theta_d, g_D, \eta_D, \beta_{D1}, \beta_{D2})$
**end for**
**Output:** $G$

---

Informally, our algorithm trains a GAN by setting the "fake" component to be the $LHS$ (in our formulation, the residuals of the equation) and the "real" component to be the $RHS$ of the equation.

This results in a GAN that learns to produce solutions that make $LHS$ indistinguishable from $RHS$, thereby approximately solving the differential equation.

## 4.1   Instance Noise

While GANs have achieved state of the art results on a wide range of generative modeling tasks, they are often difficult to train. As a result, much recent work on GANs has been dedicated to improving their sensitivity to hyperparameters and training stability [1, 2, 5, 18, 25, 28, 38, 39, 46, 49]. In our experiments, we found that DEQGAN could also be sensitive to hyperparameters, such as the Adam optimizer parameters shown in Algorithm 1.

Sønderby et al. [49] note that the convergence of GANs relies on the existence of a unique optimal discriminator that separates the distribution of "fake" samples $p_{\text{fake}}$ produced by the generator, and the distribution of the "real" data $p_{\text{data}}$. In practice, however, there may be many near-optimal discriminators that pass very different gradients to the generator, depending on their initialization. Arjovsky & Bottou [1] proved that this problem will arise when there is insufficient overlap between the supports of $p_{\text{fake}}$ and $p_{\text{data}}$. In the DEQGAN training algorithm, setting $RHS = 0$ constrains $p_{\text{data}}$ to the Dirac delta function $\delta(0)$, and therefore the distribution of "real" data to a zero-dimensional manifold. This makes it unlikely that $p_{\text{fake}}$ and $p_{\text{data}}$ will share support in a high-dimensional space.

The solution proposed by [1, 49] is to add "instance noise" to $p_{\text{fake}}$ and $p_{\text{data}}$ to encourage their overlap. This amounts to adding noise to the $LHS$ and the $RHS$, respectively, at each iteration of Algorithm 1. Because this makes the discriminator's job more difficult, we add Gaussian noise with standard deviation equal to the difference between the generator and discriminator losses, $L_g$ and $L_d$, i.e.

$$\varepsilon = \mathcal{N}(0, \sigma^2), \quad \sigma = \text{ReLU}(L_g - L_d) \tag{9}$$

As the generator and discriminator reach equilibrium, Equation 9 will naturally converge to zero. We use the ReLU function because $L_d > L_g$ indicates that the discriminator is generally performing worse than the generator, suggesting that additional noise should not be used. In Section 5.2, we conduct an ablation study and find that this improves the ability of DEQGAN to produce accurate solutions across a range of hyperparameter settings.

## 4.2   Residual Monitoring

One of the attractive properties of Algorithm 1 is that the "fake" $LHS$ vector of equation residuals gives a direct measure of solution quality at each training iteration. We observe that when DEQGAN training becomes unstable, the $LHS$ tends to oscillate wildly, while it decreases steadily throughout training for successful runs. By monitoring the $L_1$ norm of the $LHS$ in the first 25% of training iterations, we are able to easily detect and terminate poor-performing runs if the variance of these values exceeds some threshold. We provide further details on this method in Appendix A.7 and experimentally demonstrate that it is able to distinguish between DEQGAN runs that end in high and low mean squared errors in Section 5.2.

# 5   Experiments

We conducted experiments on a suite of twelve differential equations (Table 1), including highly nonlinear PDEs and systems of ODEs, comparing DEQGAN to classical unsupervised PINNs that use (squared) $L_2$, $L_1$, and Huber [24] loss functions. We also report results obtained by the fourth-order Runge-Kutta (RK4) and second-order finite difference (FD) numerical methods for initial and boundary value problems, respectively. The numerical solutions were computed over meshes containing the same number of points that were used to train the neural network methods. Details for each experiment, including exact problem specifications and hyperparameters, are provided in Appendix A.2 and A.5.

## 5.1   DEQGAN vs. Classical PINNs

We report the mean squared error of the solution obtained by each method, computed against known solutions obtained either analytically or with high-quality numerical solvers [6, 54]. We added residual connections between neighboring layers of all models, applied spectral normalization

Table 1: Summary of Experiments

| Key | Equation | Class | Order | Linear |
|---|---|---|---|---|
| EXP | $\dot{x}(t) + x(t) = 0$ | ODE | 1st | Yes |
| SHO | $\ddot{x}(t) + x(t) = 0$ | ODE | 2nd | Yes |
| NLO | $\ddot{x}(t) + 2\beta\dot{x}(t) + \omega^2 x(t) + \phi x(t)^2 + \epsilon x(t)^3 = 0$ | ODE | 2nd | No |
| COO | $\begin{cases} \dot{x}(t) = -ty \\ \dot{y}(t) = tx \end{cases}$ | ODE | 1st | Yes |
| SIR | $\begin{cases} \dot{S}(t) = -\beta I(t)S(t)/N \\ \dot{I}(t) = \beta I(t)S(t)/N - \gamma I(t) \\ \dot{R}(t) = \gamma I(t) \end{cases}$ | ODE | 1st | No |
| HAM | $\begin{cases} \dot{x}(t) = p_x \\ \dot{y}(t) = p_y \\ \dot{p_x}(t) = -V_x \\ \dot{p_y}(t) = -V_y \end{cases}$ | ODE | 1st | No |
| EIN | $\begin{cases} \dot{x}(z) = \frac{1}{z+1}(-\Omega - 2v + x + 4y + xv + x^2) \\ \dot{y}(z) = \frac{-1}{z+1}(vx\Gamma(r) - xy + 4y - 2yv) \\ \dot{v}(z) = \frac{-v}{z+1}(x\Gamma(r) + 4 - 2v) \\ \dot{\Omega}(z) = \frac{\Omega}{z+1}(-1 + 2v + x) \\ \dot{r}(z) = \frac{-r\Gamma(r)x}{z+1} \end{cases}$ | ODE | 1st | No |
| POS | $u_{xx} + u_{yy} = 2x(y-1)(y - 2x + xy + 2)e^{x-y}$ | PDE | 2nd | Yes |
| HEA | $u_t = \kappa u_{xx}$ | PDE | 2nd | Yes |
| WAV | $u_{tt} = c^2 u_{xx}$ | PDE | 2nd | Yes |
| BUR | $u_t + uu_x - \nu u_{xx} = 0$ | PDE | 2nd | No |
| ACA | $u_t - \epsilon u_{xx} - u + u^3 = 0$ | PDE | 2nd | No |

Table 2: Experimental Results

| Key | Mean Squared Error | | | | |
| | $L_1$ | $L_2$ | Huber | DEQGAN | Numerical |
|---|---|---|---|---|---|
| EXP | $3 \cdot 10^{-3}$ | $2 \cdot 10^{-5}$ | $1 \cdot 10^{-5}$ | $3 \cdot 10^{-16}$ | $2 \cdot 10^{-14}$ (RK4) |
| SHO | $9 \cdot 10^{-6}$ | $1 \cdot 10^{-10}$ | $6 \cdot 10^{-11}$ | $4 \cdot 10^{-13}$ | $1 \cdot 10^{-11}$ (RK4) |
| NLO | $6 \cdot 10^{-2}$ | $1 \cdot 10^{-9}$ | $9 \cdot 10^{-10}$ | $1 \cdot 10^{-12}$ | $4 \cdot 10^{-11}$ (RK4) |
| COO | $5 \cdot 10^{-1}$ | $1 \cdot 10^{-7}$ | $1 \cdot 10^{-7}$ | $1 \cdot 10^{-8}$ | $2 \cdot 10^{-9}$ (RK4) |
| SIR | $7 \cdot 10^{-5}$ | $3 \cdot 10^{-9}$ | $1 \cdot 10^{-9}$ | $1 \cdot 10^{-10}$ | $5 \cdot 10^{-13}$ (RK4) |
| HAM | $1 \cdot 10^{-1}$ | $2 \cdot 10^{-7}$ | $9 \cdot 10^{-8}$ | $1 \cdot 10^{-10}$ | $7 \cdot 10^{-14}$ (RK4) |
| EIN | $6 \cdot 10^{-2}$ | $2 \cdot 10^{-2}$ | $1 \cdot 10^{-2}$ | $3 \cdot 10^{-4}$ | $4 \cdot 10^{-7}$ (RK4) |
| POS | $4 \cdot 10^{-6}$ | $1 \cdot 10^{-10}$ | $6 \cdot 10^{-11}$ | $4 \cdot 10^{-13}$ | $3 \cdot 10^{-10}$ (FD) |
| HEA | $6 \cdot 10^{-3}$ | $3 \cdot 10^{-5}$ | $1 \cdot 10^{-5}$ | $6 \cdot 10^{-10}$ | $4 \cdot 10^{-7}$ (FD) |
| WAV | $6 \cdot 10^{-2}$ | $4 \cdot 10^{-5}$ | $6 \cdot 10^{-4}$ | $1 \cdot 10^{-8}$ | $7 \cdot 10^{-5}$ (FD) |
| BUR | $4 \cdot 10^{-3}$ | $2 \cdot 10^{-4}$ | $1 \cdot 10^{-4}$ | $4 \cdot 10^{-6}$ | $1 \cdot 10^{-3}$ (FD) |
| ACA | $6 \cdot 10^{-2}$ | $9 \cdot 10^{-3}$ | $4 \cdot 10^{-3}$ | $3 \cdot 10^{-3}$ | $2 \cdot 10^{-4}$ (FD) |

to the discriminator, added instance noise to the $p_{\text{fake}}$ and $p_{\text{real}}$, and used residual monitoring to terminate poor-performing runs in the first 25% of training iterations. Results were obtained with hyperparameters tuned for DEQGAN. In Appendix A.6, we tuned each classical PINN method for comparison, but did not observe a significant difference.

Table 2 reports the lowest mean squared error obtained by each method across ten different model weight initializations. We see that DEQGAN obtains lower mean squared errors than classical PINNs that use $L_2$, $L_1$, and Huber loss functions for all twelve problems, often by several orders of magnitude. DEQGAN also achieves solution accuracies that are competitive with the RK4 and FD numerical methods.

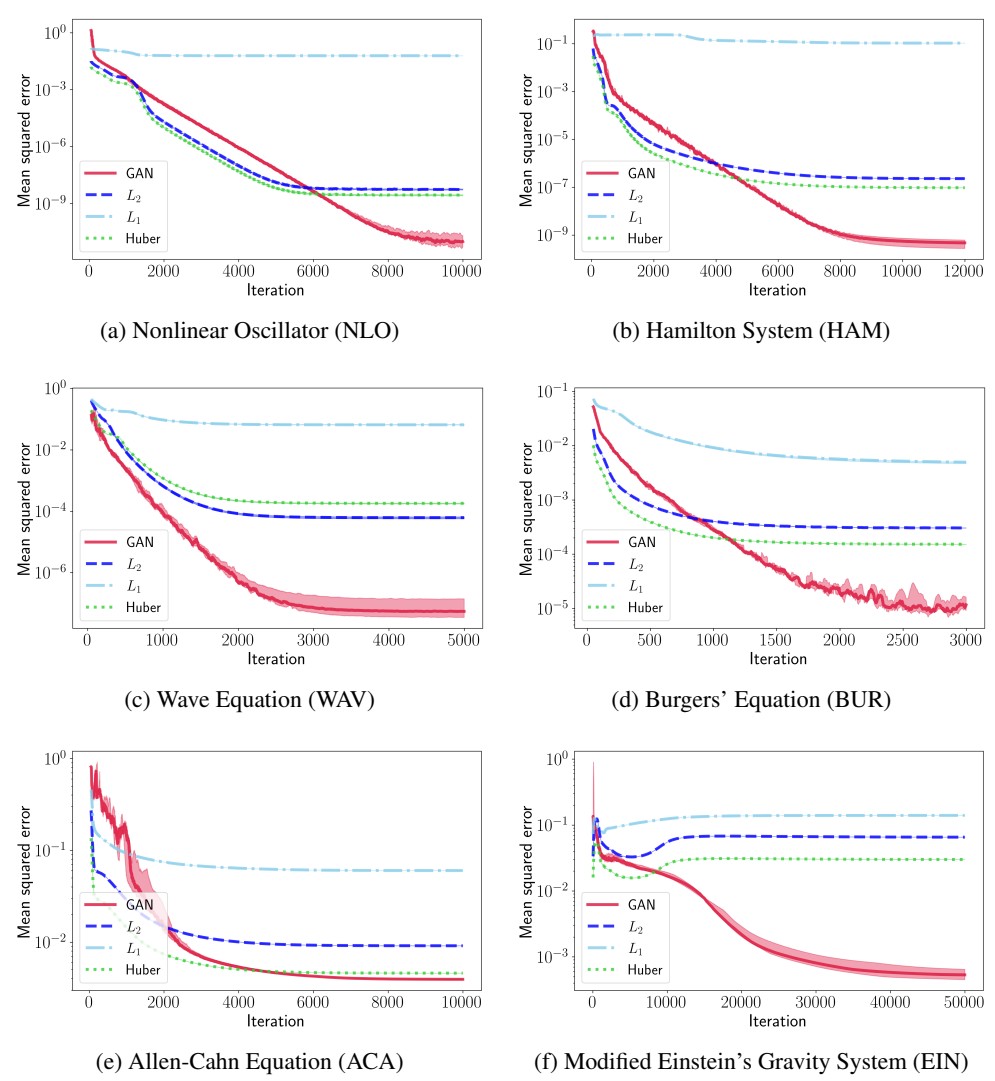

(a) Nonlinear Oscillator (NLO)

(b) Hamilton System (HAM)

(c) Wave Equation (WAV)

(d) Burgers' Equation (BUR)

(e) Allen-Cahn Equation (ACA)

(f) Modified Einstein's Gravity System (EIN)

Figure 2: Mean squared errors vs. iteration for DEQGAN, $L_2$, $L_1$, and Huber loss for six equations. We perform ten randomized trials and plot the median (bold) and $(25, 75)$ percentile range (shaded). We smooth the values using a simple moving average with window size 50.

Figure 2 plots the mean squared error vs. training iteration for six challenging equations and highlights multiple advantages of using DEQGAN over a pre-specified loss function (equivalent plots for the other six problems are provided in Appendix A.3). In particular, there is considerable variation in the quality of the solutions obtained by the classical PINNs. For example, while Huber performs better than $L_2$ on the Allen-Cahn PDE, it is outperformed by $L_2$ on the wave equation. Furthermore, Figure 2f shows that the $L_2$, $L_1$ and Huber losses all fail to converge to an accurate solution to the modified Einstein's gravity equations. Although this system has previously been solved using PINNs, the networks relied on a custom loss function that incorporated equation-specific parameters [7]. DEQGAN, however, is able to *automatically* learn a loss function that optimizes the generator to produce accurate solutions. DEQGAN solutions to four example equations are visualized in Figure 4, which shows that the ODE solutions are indistinguishable from those obtained using a numerical integrator. Similar plots for the other experiments are provided in Appendix A.2.

## 5.2 DEQGAN Training Stability: Ablation Study

In our experiments, we used instance noise to adaptively improve the training convergence of DEQGAN and employed residual monitoring to terminate poor-performing runs early. To quantify

the increased robustness offered by these techniques, we performed an ablation study comparing the percentage of high MSE ($\geq 10^{-5}$) runs obtained by 500 randomized DEQGAN runs on the exponential decay equation.

Figure 3 plots the results of these 500 DEQGAN experiments with instance noise added. For each experiment, we uniformly selected a random seed controlling model weight initialization as an integer from the range $[0, 9]$, as well as separate learning rates for the discriminator and generator in the range $[0.01, 0.1]$. We then recorded the final mean squared error after running DEQGAN training for 1000 iterations. The red lines represent runs which would be terminated early by our residual monitoring method, while the blue lines represent those which would be run to completion. We see that the large majority of hyperparameter settings tested with the addition of instance noise resulted in low mean squared errors. Further, residual monitoring was able to detect all runs with MSE $\geq 10^{-5}$. Approximately half of the MSE runs in $[10^{-8}, 10^{-5}]$ would be terminated, while 96% of runs with MSE $\leq 10^{-8}$ would be run to completion.

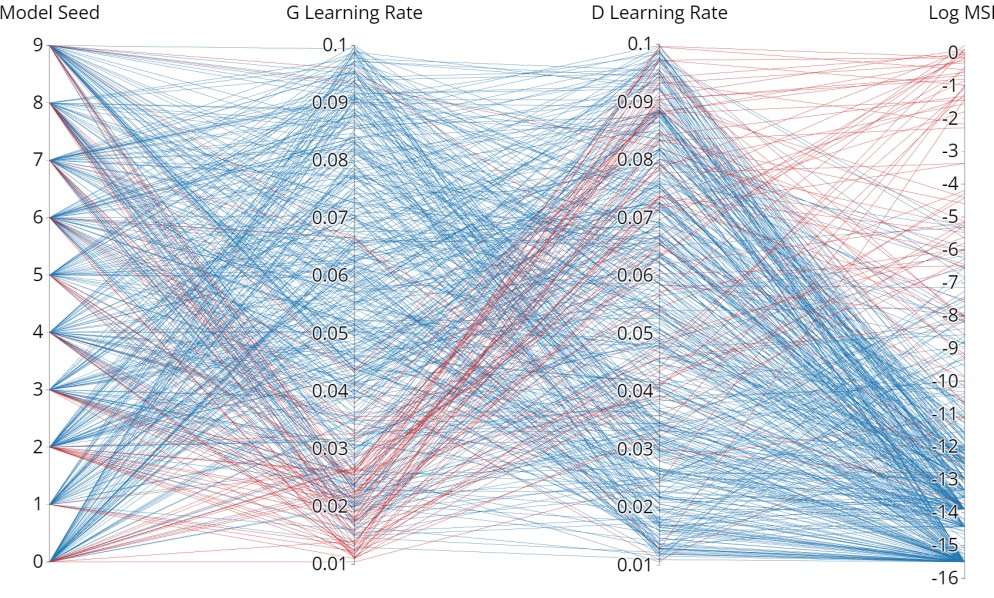

Figure 3: Parallel plot showing the results of 500 DEQGAN experiments on the exponential decay equation with instance noise. The red lines represent runs which would be terminated early by monitoring the variance of the equation residuals in the first 25% of training iterations. The mean squared error is plotted on a $log_{10}$ scale.

Table 3: Ablation Study Results

|  | % Runs with High MSE ($\geq 10^{-5}$) | |
|  | No Residual Monitoring | With Residual Monitoring |
| No Instance Noise | 12.4 | 0.4 |
| With Instance Noise | 8.0 | 0.0 |

Table 3 compares the percentage of high MSE runs with and without instance noise and residual monitoring. We see that adding instance noise decreased the percentage of runs with high MSE and that residual monitoring is highly effective at filtering out poor performing runs. When used together, these techniques eliminated all runs with MSE $\geq 10^{-5}$. These results agree with previous works, which have found that instance noise can improve the convergence of other GAN training algorithms [1, 49]. Further, they suggest that residual monitoring provides a useful performance metric that could be applied to other PINN methods for solving differential equations.

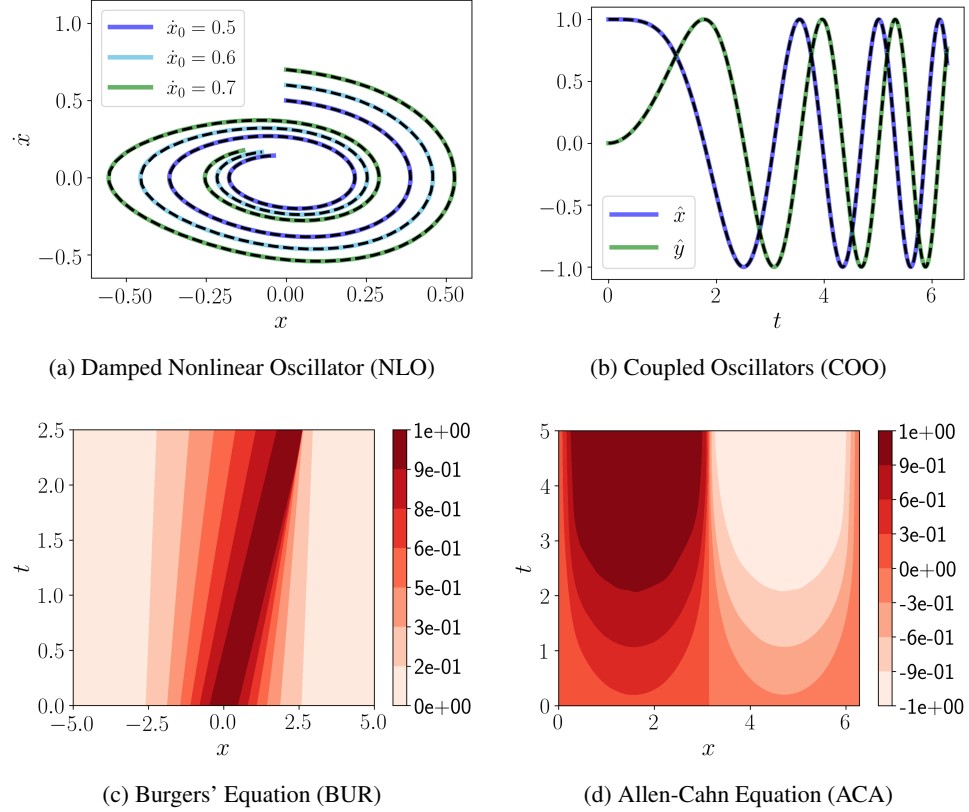

(a) Damped Nonlinear Oscillator (NLO)

(b) Coupled Oscillators (COO)

(c) Burgers' Equation (BUR)

(d) Allen-Cahn Equation (ACA)

Figure 4: Visualization of DEQGAN solutions to four equations. The top left figure plots the phase space of the DEQGAN solutions (solid color lines) obtained for three initial conditions on the NLO problem, which is solved as a second-order ODE, and known solutions computed by a numerical integrator (dashed black lines). The figure to the right plots the DEQGAN solution to the COO problem, which is solved as a system of two first-order ODEs. The second row shows contour plots of the solutions obtained by DEQGAN on the BUR and ACA problems, both nonlinear PDEs.

## 6   Conclusion

PINNs offer a promising approach to solving differential equations and to applying deep learning methods to challenging problems in science and engineering. Classical PINNs, however, lack a theoretical justification for the use of any particular loss function. In this work, we presented Differential Equation GAN (DEQGAN), a novel method that leverages GAN-based adversarial training to "learn" the loss function for solving differential equations with PINNs. We demonstrated the advantage of this approach in comparison to using classical PINNs with pre-specified loss functions, which showed varied performance and failed to converge to an accurate solution to the modified Einstein's gravity equations. In general, we demonstrated that our method can obtain multiple orders of magnitude lower mean squared errors than PINNs that use $L_2$, $L_1$ and Huber loss functions, including on highly nonlinear PDEs and systems of ODEs. Further, we showed that DEQGAN achieves solution accuracies that are competitive with the fourth-order Runge Kutta and second-order finite difference numerical methods. Finally, we found that instance noise improved training stability and that residual monitoring provides a useful performance metric for PINNs. While the equation residuals are a good measure of solution quality, PINNs lack the error bounds enjoyed by numerical methods. Formalizing these bounds is an interesting avenue for future work and would enable PINNs to be more safely deployed in real-world applications. Further, while our results evidence the advantage of "learning the loss function" with a GAN, understanding exactly what the discriminator learns is an open problem. Post-hoc explainability methods, for example, might provide useful tools for characterizing the differences between classical losses and the loss functions learned by DEQGAN, which could deepen our understanding of PINN optimization more generally.

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

# A  Appendix

## A.1  Classical Loss Functions

A plot of the various classical loss functions is provided in Figure 5.

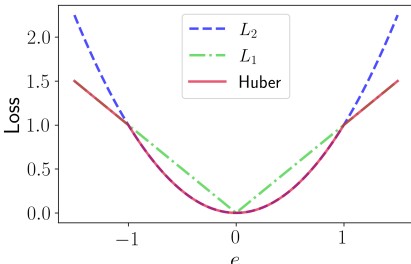

Figure 5: Comparison of $L_2$, $L_1$, and Huber loss functions. The Huber loss is equal to $L_2$ for $e \leq 1$ and to $L_1$ for $e > 1$.

## A.2  Description of Experiments

### A.2.1  Exponential Decay (EXP)

Consider a model for population decay $x(t)$ given by the exponential differential equation

$$\dot{x}(t) + x(t) = 0, \tag{10}$$

with $x(0) = 1$ and $t \in [0, 10]$. The ground truth solution $x(t) = e^{-t}$ can be obtained analytically, which we use to calculate the mean squared error of the predicted solution.

To set up the problem for DEQGAN, we define $LHS = \dot{x} + x$ and $RHS = 0$. Figure 6 presents the results from training DEQGAN on this equation.

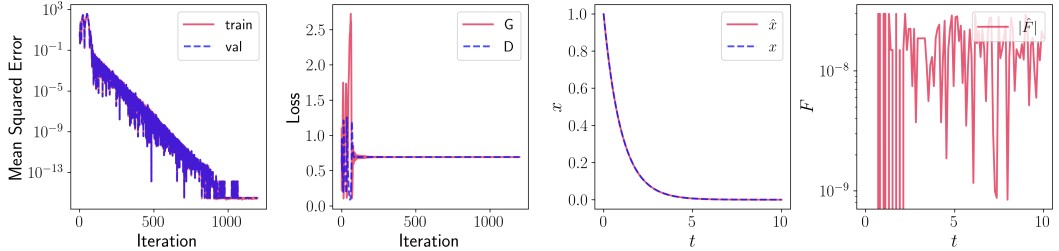

Figure 6: Visualization of DEQGAN training for the exponential decay problem. The left-most figure plots the mean squared error vs. iteration. To the right, we plot the value of the generator (G) and discriminator (D) losses at each iteration. Right of this we plot the prediction of the generator $\hat{x}$ and the true analytic solution $x$ as functions of time $t$. The right-most figure plots the absolute value of the residual of the predicted solution $\hat{F}$.

### A.2.2  Simple Harmonic Oscillator (SHO)

Consider the motion of an oscillating body $x(t)$, which can be modeled by the simple harmonic oscillator differential equation

$$\ddot{x}(t) + x(t) = 0, \tag{11}$$

with $x(0) = 0$, $\dot{x}(0) = 1$, and $t \in [0, 2\pi]$. This differential equation can be solved analytically and has an exact solution $x(t) = \sin t$.

Here we set $LHS = \ddot{x} + x$ and $RHS = 0$. Figure 7 plots the results of training DEQGAN on this problem.

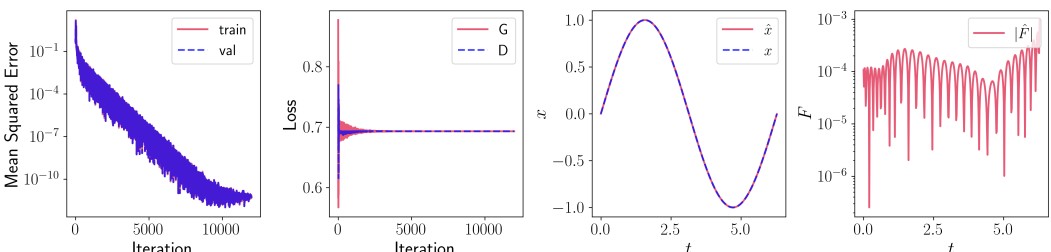

Figure 7: Visualization of DEQGAN training for the simple harmonic oscillator problem.

### A.2.3 Damped Nonlinear Oscillator (NLO)

Further increasing the complexity of the differential equations being considered, consider a less idealized oscillating body subject to additional forces, whose motion $x(t)$ we can described by the nonlinear oscillator differential equation

$$\ddot{x}(t) + 2\beta\dot{x}(t) + \omega^2 x(t) + \phi x(t)^2 + \epsilon x(t)^3 = 0, \tag{12}$$

with $\beta = 0.1, \omega = 1, \phi = 1, \epsilon = 0.1$, $x(0) = 0$, $\dot{x}(0) = 0.5$, and $t \in [0, 4\pi]$. This equation does not admit an analytical solution. Instead, we use the high-quality solver provided by SciPy's `solve_ivp` [54].

We set $LHS = \ddot{x} + 2\beta\dot{x} + \omega^2 x + \phi x^2 + \epsilon x^3 = 0$ and $RHS = 0$. Figure 8 plots the results obtained from training DEQGAN on this equation.

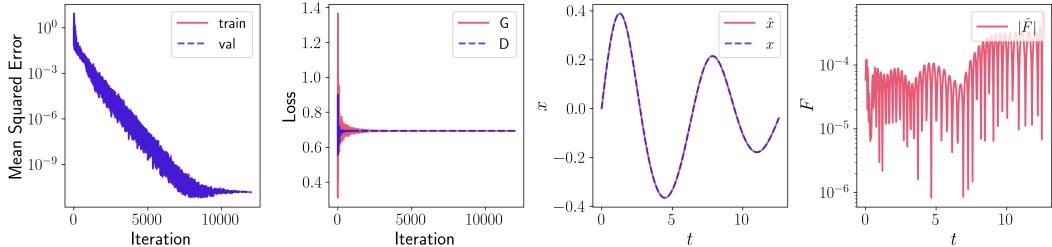

Figure 8: Visualization of DEQGAN training for the nonlinear oscillator problem.

### A.2.4 Coupled Oscillators (COO)

Consider the system of ordinary differential equations given by

$$\begin{cases} \dot{x}(t) = -ty \\ \dot{y}(t) = tx \end{cases} \tag{13}$$

with $x(0) = 1$, $y(0) = 0$, and $t \in [0, 2\pi]$. This equation has an exact analytical solution given by

$$\begin{cases} x = \cos\left(\dfrac{t^2}{2}\right) \\ y = \sin\left(\dfrac{t^2}{2}\right) \end{cases} \tag{14}$$

Here we set

$$LHS = \left[\frac{dx}{dt} + ty, \frac{dy}{dt} - xy\right]^T \tag{15}$$

and $RHS = [0, 0]^T$. Figure 9 plots the result of training DEQGAN on this problem.

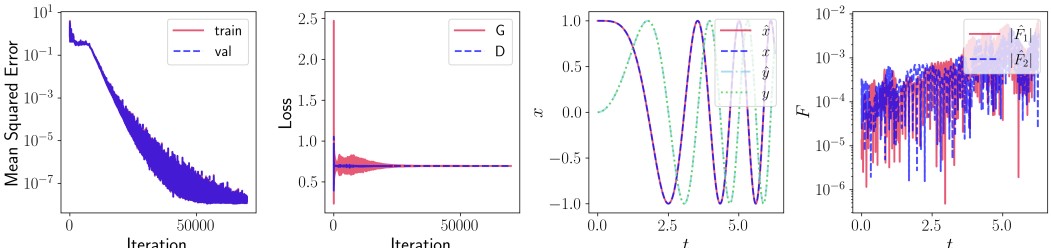

Figure 9: Visualization of DEQGAN training for the coupled oscillators system of equations. In the third figure, we plot the predictions of the generator $\hat{x}, \hat{y}$ and the true analytic solutions $x, y$ as functions of time $t$. The right-most figure plots the absolute value of the residuals of the predicted solution $\hat{F}_j$ for each equation $j$.

### A.2.5 SIR Epidemiological Model (SIR)

Given the ongoing pandemic of novel coronavirus (COVID-19) [55], we consider an epidemiological model of infectious disease spread given by a system of ordinary differential equations. Specifically, consider the Susceptible $S(t)$, Infected $I(t)$, Recovered $R(t)$ model for the spread of an infectious disease over time $t$. The model is defined by a system of three ordinary differential equations

$$
\begin{cases}
\dot{S}(t) = -\beta \dfrac{IS}{N} \\[2mm]
\dot{I}(t) = \beta \dfrac{IS}{N} - \gamma I \\[2mm]
\dot{R}(t) = \gamma I
\end{cases}
\tag{16}
$$

where $\beta = 3, \gamma = 1$ are given constants related to the infectiousness of the disease, $N = S + I + R$ is the (constant) total population, $S(0) = 0.99, I(0) = 0.01, R(0) = 0$, and $t \in [0, 10]$. As this system has no analytical solution, we use SciPy's `solve_ivp` solver [54] to obtain ground truth solutions.

We set $LHS$ to be the vector

$$
LHS = \left[ \frac{dS}{dt} + \beta \frac{IS}{N}, \frac{dI}{dt} - \beta \frac{IS}{N} + \gamma I, \frac{dR}{dt} - \gamma I \right]^T
\tag{17}
$$

and $RHS = [0, 0, 0]^T$. We present the results of training DEQGAN to solve this system of differential equations in Figure 10.

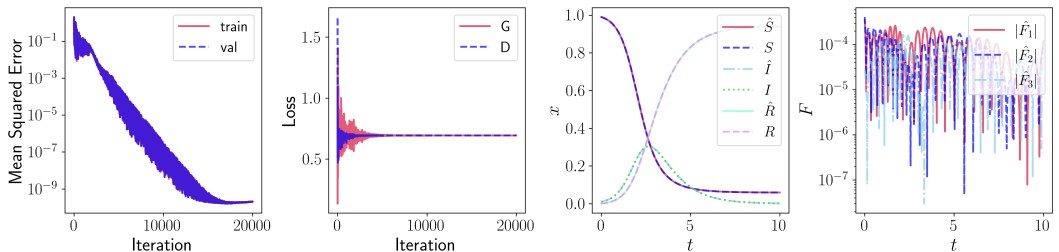

Figure 10: Visualization of DEQGAN training for the SIR system of equations.

### A.2.6 Hamiltonian System (HAM)

Consider a particle moving through a potential $V$, the trajectory of which is described by the system of ordinary differential equations

$$\begin{cases} \dot{x}(t) = p_x \\ \dot{y}(t) = p_y \\ \dot{p_x}(t) = -V_x \\ \dot{p_y}(t) = -V_y \end{cases} \tag{18}$$

with $x(0) = 0, y(0) = 0.3, p_x(0) = 1, p_y(0) = 0$, and $t \in [0, 1]$. $V_x$ and $V_y$ are the $x$ and $y$ derivatives of the potential $V$, which we construct by summing ten random bivariate Gaussians

$$V = -\frac{A}{2\pi\sigma^2} \sum_{i=1}^{10} \exp\left(-\frac{1}{2\sigma^2}||\mathbf{x}(t) - \mu_i||_2^2\right) \tag{19}$$

where $\mathbf{x}(t) = [x(t), y(t)]^T$, $A = 0.1, \sigma = 0.1$, and each $\mu_i$ is sampled from $[0, 1] \times [0, 1]$ uniformly at random. As before, we use SciPy to obtain ground-truth solutions.

We set $LHS$ to be the vector

$$LHS = \left[\frac{dx}{dt} - p_x, \frac{dy}{dt} - p_y, \frac{dp_x}{dt} + V_x, \frac{dp_y}{dt} + V_y\right]^T \tag{20}$$

and $RHS = [0, 0, 0, 0]^T$. We present the results of training DEQGAN to solve this system of differential equations in Figure 11.

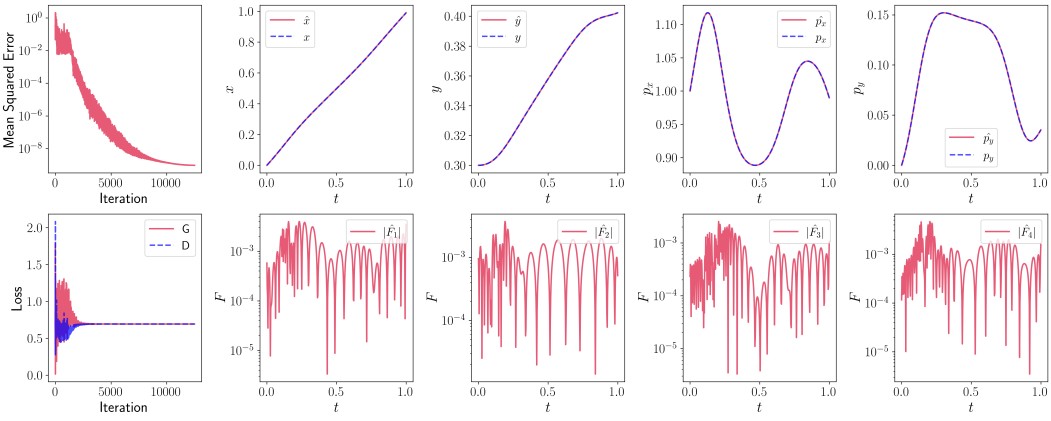

Figure 11: Visualization of DEQGAN training for the Hamiltonian system of equations. For ease of visualization, we plot the predictions and residuals for each equation separately.

### A.2.7 Modified Einstein's Gravity System (EIN)

The most challenging system of ODEs we consider comes from Einstein's theory of general relativity. Following observations from type Ia supernovae in 1998 [45], several cosmological models have been proposed to explain the accelerated expansion of the universe. Some of these rely on the existence of unobserved forms such as dark energy and dark matter, while others directly modify Einstein's theory.

Hu-Sawicky $f(R)$ gravity is one model that falls under this category. Chantada et al. [7] show how the following system of five ODEs can be derived from the modified field equations implied by this model.

$$\begin{cases} \dot{x}(z) = \dfrac{1}{z+1}(-\Omega - 2v + x + 4y + xv + x^2) \\[2mm] \dot{y}(z) = \dfrac{-1}{z+1}(vx\Gamma(r) - xy + 4y - 2yv) \\[2mm] \dot{v}(z) = \dfrac{-v}{z+1}(x\Gamma(r) + 4 - 2v) \\[2mm] \dot{\Omega}(z) = \dfrac{\Omega}{z+1}(-1 + 2v + x) \\[2mm] \dot{r}(z) = \dfrac{-r\Gamma(r)x}{z+1} \end{cases} \tag{21}$$

where

$$\Gamma(r) = \frac{(r+b)\left[(r+b)^2 - 2b\right]}{4br}. \tag{22}$$

The initial conditions are given by

$$\begin{cases} x_0 = 0 \\[2mm] y_0 = \dfrac{\Omega_{m,0}(1+z_0)^3 + 2(1-\Omega_{m,0})}{2\left[\Omega_{m,0}(1+z_0)^3 + (1-\Omega_{m,0})\right]} \\[2mm] v_0 = \dfrac{\Omega_{m,0}(1+z_0)^3 + 4(1-\Omega_{m,0})}{2\left[\Omega_{m,0}(1+z_0)^3 + (1-\Omega_{m,0})\right]} \\[2mm] \Omega_0 = \dfrac{\Omega_{m,0}(1+z_0)^3}{\Omega_{m,0}(1+z_0)^3 + (1-\Omega_{m,0})} \\[2mm] r_0 = \dfrac{\Omega_{m,0}(1+z_0)^3 + 4(1-\Omega_{m,0})}{(1-\Omega_{m,0})} \end{cases} \tag{23}$$

where $z_0 = 10, \Omega_{m,0} = 0.15, b = 5$ and we solve the system for $z \in [0, z_0]$. While the physical interpretation of the various parameters is beyond the scope of this paper, we note that Equations 21 and 22 exhibit a high degree of non-linearity. Ground truth solutions are again obtained using SciPy, and the results obtained by DEQGAN are shown in Figure 12.

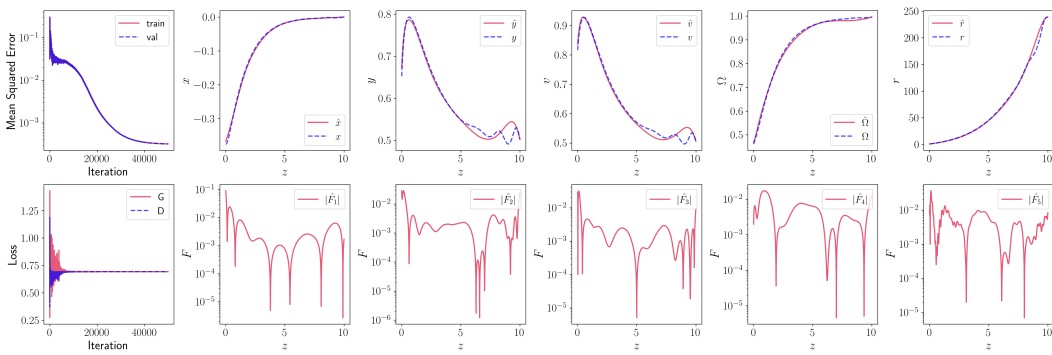

Figure 12: Visualization of DEQGAN training for the modified Einstein's gravity system of equations. For ease of visualization, we plot the predictions and residuals for each equation separately.

### A.2.8 Poisson Equation (POS)

Consider the Poisson partial differential equation (PDE) given by

$$\frac{\partial^2 u}{\partial x^2} + \frac{\partial^2 u}{\partial y^2} = 2x(y-1)(y-2x+xy+2)e^{x-y} \tag{24}$$

where $(x, y) \in [0, 1] \times [0, 1]$. The equation is subject to Dirichlet boundary conditions on the edges of the unit square

$$
\begin{aligned}
u(x, y)\Big|_{x=0} &= 0 \\
u(x, y)\Big|_{x=1} &= 0 \\
u(x, y)\Big|_{y=0} &= 0 \\
u(x, y)\Big|_{y=1} &= 0.
\end{aligned}
\tag{25}
$$

The analytical solution is

$$
u(x, y) = x(1-x)y(1-y)e^{x-y}.
\tag{26}
$$

We use the two-dimensional Dirichlet boundary adjustment formulae provided in Chen et al. [8]. To set up the problem for DEQGAN we let

$$
LHS = \frac{\partial^2 u}{\partial x^2} + \frac{\partial^2 u}{\partial y^2} - 2x(y-1)(y-2x+xy+2)e^{x-y}
\tag{27}
$$

and $RHS = 0$. We present the results of training DEQGAN on this problem in Figure 13.

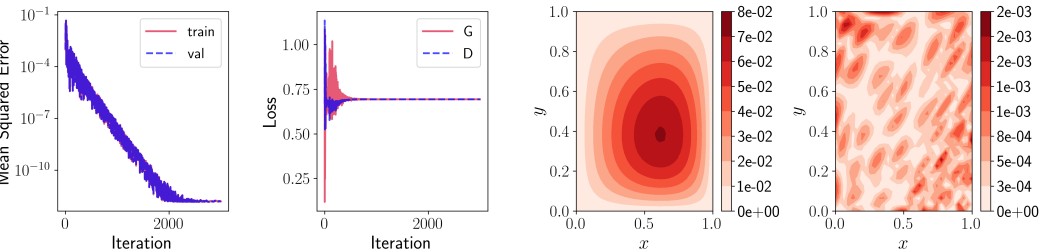

Figure 13: Visualization of DEQGAN training for the Poisson equation. In the third figure, we plot the prediction of the generator $\hat{u}$ as a function of position $(x, y)$. The right-most figure plots the absolute value of the residual $\hat{F}$, as a function of $(x, y)$.

### A.2.9 Heat Equation (HEA)

We consider the time-dependent heat (diffusion) equation given by

$$
\frac{\partial u}{\partial t} = \kappa \frac{\partial^2 u}{\partial x^2}
\tag{28}
$$

where $\kappa = 1$ and $(x, t) \in [0, 1] \times [0, 0.2]$. The equation is subject to an initial condition and Dirichlet boundary conditions given by

$$
\begin{aligned}
u(x, y)\Big|_{t=0} &= \sin(\pi x) \\
u(x, y)\Big|_{x=0} &= 0 \\
u(x, y)\Big|_{x=1} &= 0
\end{aligned}
\tag{29}
$$

and has an analytical solution

$$
u(x, y) = e^{-\kappa \pi^2 t} \sin(\pi x).
\tag{30}
$$

The results obtained by DEQGAN on this problem are shown in Figure 14.

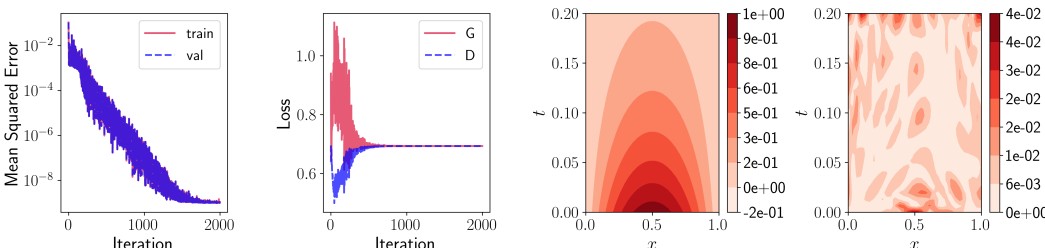

Figure 14: Visualization of DEQGAN training for the heat equation. In the third figure, we plot the prediction of the generator $\hat{u}$ as a function of position $(x, t)$. The right-most figure plots the absolute value of the residual $\hat{F}$, as a function of $(x, t)$.

### A.2.10 Wave Equation (WAV)

Consider the time-dependent wave equation given by

$$\frac{\partial^2 u}{\partial t^2} = c^2 \frac{\partial^2 u}{\partial x^2} \tag{31}$$

where $c = 1$ and $(x, t) \in [0, 1] \times [0, 1]$. This formulation is very similar to the heat equation but involves a second order derivative with respect to time. We subject the equation to the same initial condition and boundary conditions as 29 but require an added Neumann condition due to the equation's second time derivative.

$$
\begin{aligned}
u(x, y)\bigg|_{t=0} &= \sin(\pi x) \\
u_t(x, y)\bigg|_{t=0} &= 0 \\
u(x, y)\bigg|_{x=0} &= 0 \\
u(x, y)\bigg|_{x=1} &= 0
\end{aligned}
\tag{32}
$$

This yields the analytical solution

$$u(x, y) = \cos(c\pi t)\sin(\pi x). \tag{33}$$

The results of training DEQGAN on this problem are shown in Figure 14.

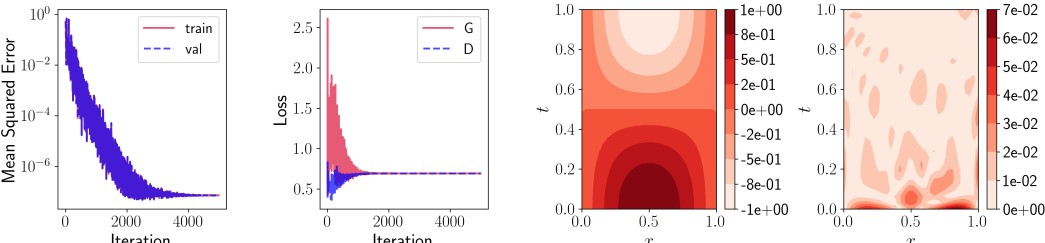

Figure 15: Visualization of DEQGAN training for the wave equation.

### A.2.11 Bugers' Equation (BUR)

Moving to non-linear PDEs, we consider the viscous Burgers' equation given by

$$\frac{\partial u}{\partial t} + u\frac{\partial u}{\partial x} = \nu\frac{\partial^2 u}{\partial x^2} \tag{34}$$

where $\nu = 0.001$ and $(x, t) \in [-5, 5] \times [0, 2.5]$. To specify the equation, we use the following initial condition and Dirichlet boundary conditions:

$$u(x, y)\Big|_{t=0} = \frac{1}{\cosh(x)}$$
$$u(x, y)\Big|_{x=-5} = 0 \quad (35)$$
$$u(x, y)\Big|_{x=5} = 0$$

As this equation has no analytical solution, we use the fast Fourier transform (FFT) method [6] to obtain ground truth solutions. The results obtained by DEQGAN are summarized by Figure 16. As time progresses, we see the formation of a "shock wave" that becomes increasingly steep but remains smooth due to the regularizing diffusive term $\nu u_{xx}$.

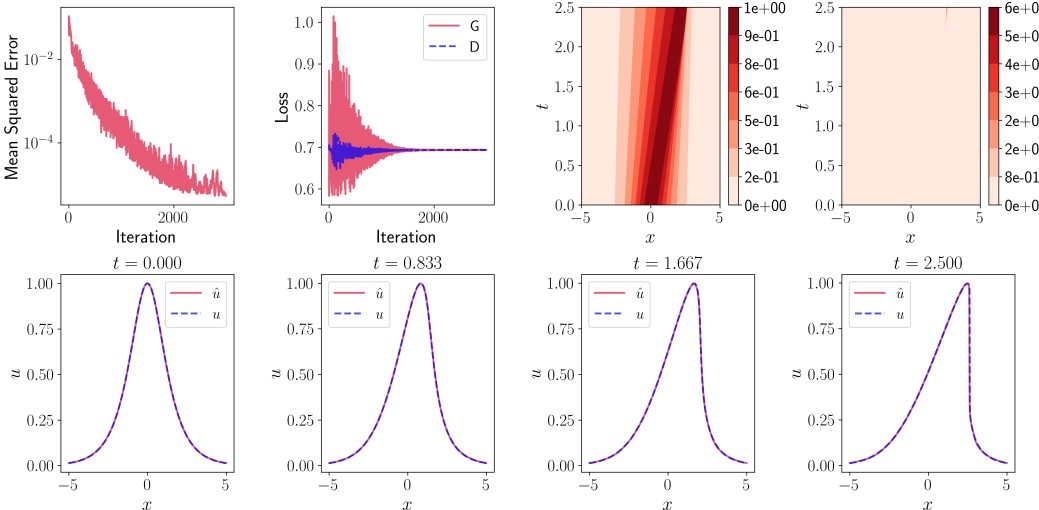

Figure 16: Visualization of DEQGAN training for Bugers' equation. The plots in the second row show "snapshots" of the 1D wave at different points along the time domain.

### A.2.12 Allen-Cahn Equation (ACA)

Finally, we consider the Allen-Cahn PDE, a well-known reaction-diffusion equation given by

$$\frac{\partial u}{\partial t} - \epsilon \frac{\partial^2 u}{\partial x^2} - u + u^3 = 0 \quad (36)$$

where $\epsilon = 0.001$ and $(x, t) \in [0, 2\pi] \times [0, 5]$. We subject the equation to an initial condition and Dirichlet boundary conditions given by

$$u(x, y)\Big|_{t=0} = \frac{1}{4}\sin(x)$$
$$u(x, y)\Big|_{x=0} = 0 \quad (37)$$
$$u(x, y)\Big|_{x=2\pi} = 0$$

The results are shown in Figure 17. We see that as time progresses, the sinusoidal initial condition transforms into a square wave, becoming very steep at the turning points of the solution.

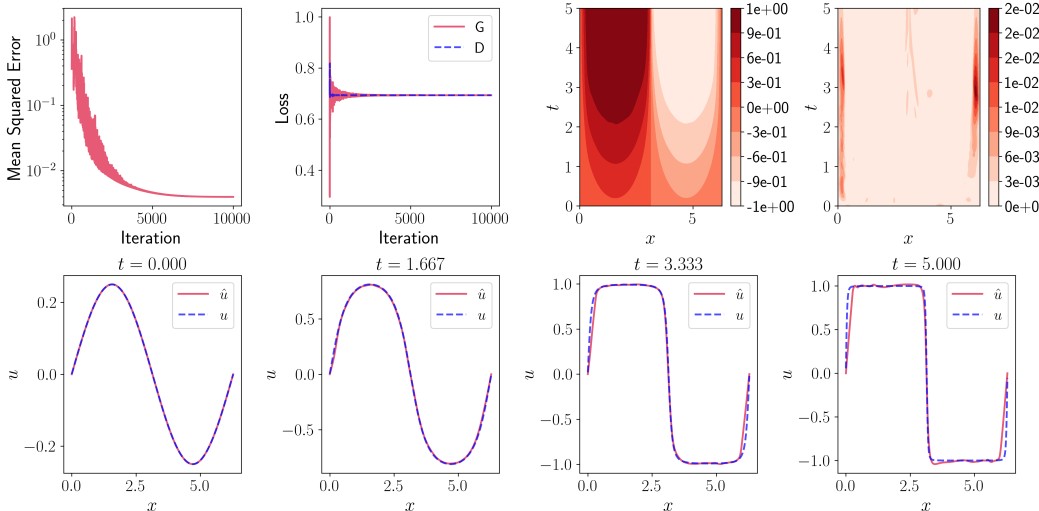

Figure 17: Visualization of DEQGAN training for the Allen-Cahn equation. The plots in the second row show "snapshots" of the 1D wave at different points along the time domain.

**A.3    Method Comparison for Other Experiments**

Figure 18 visualizes the training results achieved by DEQGAN and the alternative unsupervised neural networks that use $L_2$, $L_1$ and Huber loss functions for the remaining six problems.

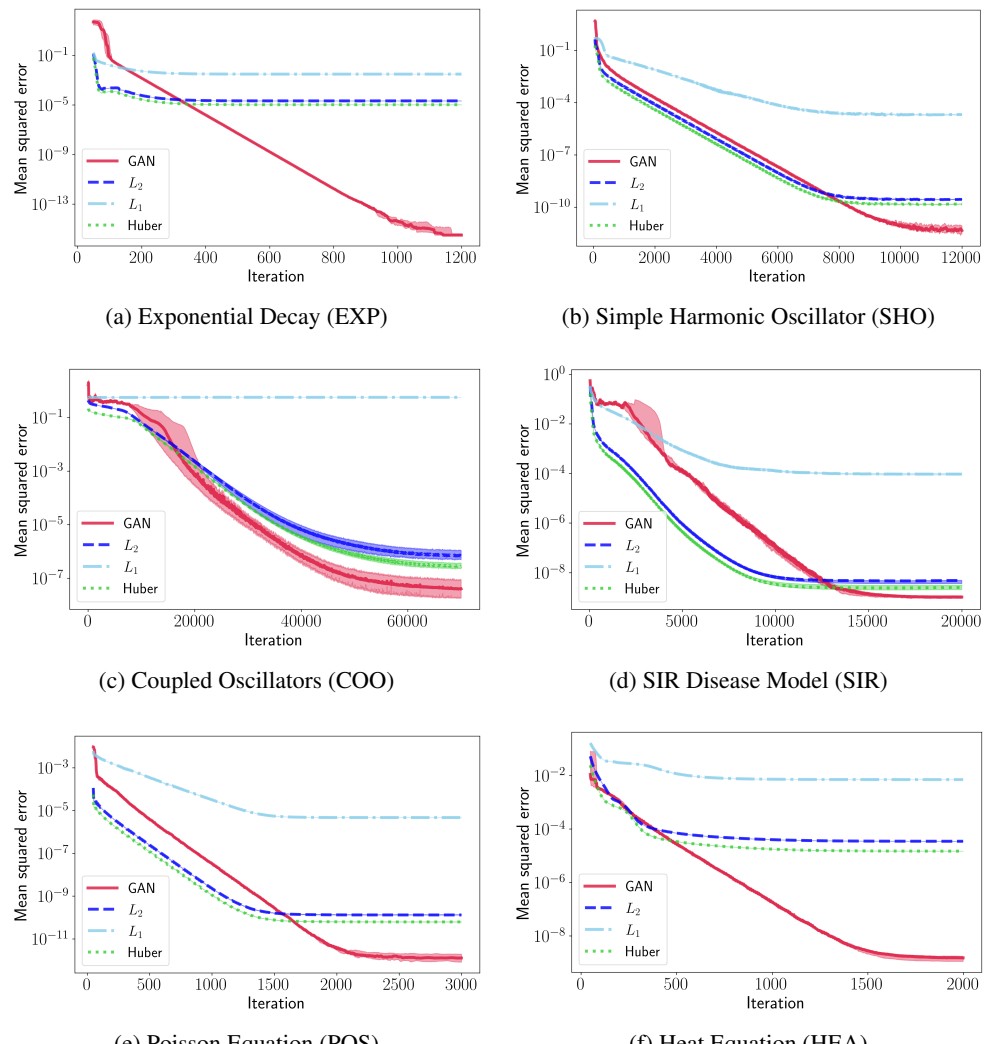

(a) Exponential Decay (EXP)         (b) Simple Harmonic Oscillator (SHO)

(c) Coupled Oscillators (COO)         (d) SIR Disease Model (SIR)

(e) Poisson Equation (POS)         (f) Heat Equation (HEA)

Figure 18: Mean squared errors vs. iteration for DEQGAN, $L_2$, $L_1$, and Huber loss for various equations. We perform ten randomized trials and plot the median (bold) and $(25, 75)$ percentile range (shaded). We smooth the values using a simple moving average with window size $50$.

**A.4    DEQGAN Training and Architecture**

**A.4.1    Two Time-Scale Update Rule**

Heusel et al. [23] proposed the two time-scale update rule (TTUR) for training GANs, a method in which the discriminator and generator are trained with separate learning rates. They showed that their method led to improved performance and proved that, in some cases, TTUR ensures convergence to a stable local Nash equilibrium. One intuition for TTUR comes from the potentially different loss surfaces of the discriminator and generator. Allowing learning rates to be tuned to a particular loss surface can enable more efficient gradient-based optimization. We make use of TTUR throughout this paper as an instrumental lever when tuning GANs to reach desired performance.

### A.4.2 Spectral Normalization

Proposed by Miyato et al. [39], Spectrally Normalized GAN (SN-GAN) is a method for controlling exploding discriminator gradients when optimizing Equation 3 that leverages a novel weight normalization technique. The key idea is to control the Lipschitz constant of the discriminator by constraining the spectral norm of each layer in the discriminator. Specifically, the authors propose dividing the weight matrices $W_i$ of each layer $i$ by their spectral norm $\sigma(W_i)$

$$W_{SN,i} = \frac{W_i}{\sigma(W_i)}, \tag{38}$$

where

$$\sigma(W_i) = \max_{\|h_i\|_2 \leq 1} \|W_i h_i\|_2 \tag{39}$$

and $h_i$ denotes the input to layer $i$. The authors prove that this normalization technique bounds the Lipschitz constant of the discriminator above by 1, thus strictly enforcing the 1-Lipshcitz constraint on the discriminator. In our experiments, adopting the SN-GAN formulation led to even better performance than WGAN-GP [2, 18].

### A.4.3 Residual Connections

He et al. [21] showed that the addition of residual connections improves deep neural network training. We employ residual connections in our networks, as they allow gradients to flow more easily through the models and thereby reduce numerical instability. Residual connections augment a typical activation with the identity operation.

$$y = \mathcal{F}(x, W_i) + x \tag{40}$$

where $\mathcal{F}$ is the activation function, $x$ is the input to the unit, $W_i$ are the weights and $y$ is the output of the unit. This acts as a "skip connection", allowing inputs and gradients to forego the nonlinear component.

### A.5 DEQGAN Hyperparameters

We used Ray Tune [33] to tune DEQGAN hyperparameters for each differential equation. Tables 4 and 5 summarize these hyperparameter values for the ODE and PDE problems, respectively. The experiments and hyperparameter tuning conducted for this research totaled 13,272 hours of compute performed on Intel Cascade Lake CPU cores belonging to an internal cluster.

Table 4: Hyperparameter Settings for DEQGAN (ODEs)

| HYPERPARAMETER | EXP | SHO | NLO | COO | SIR | HAM | EIN |
|---|---|---|---|---|---|---|---|
| NUM. ITERATIONS | 1200 | 12000 | 12000 | 70000 | 20000 | 12500 | 50000 |
| NUM. GRID POINTS | 100 | 400 | 400 | 800 | 800 | 400 | 1000 |
| $G$ UNITS/LAYER | 40 | 40 | 40 | 40 | 50 | 40 | 40 |
| $G$ NUM. LAYERS | 2 | 3 | 4 | 5 | 4 | 5 | 4 |
| $D$ UNITS/LAYER | 20 | 50 | 20 | 40 | 50 | 50 | 30 |
| $D$ NUM. LAYERS | 4 | 3 | 2 | 2 | 4 | 2 | 2 |
| ACTIVATIONS | tanh | tanh | tanh | tanh | tanh | tanh | tanh |
| $G$ LEARNING RATE | 0.094 | 0.005 | 0.010 | 0.004 | 0.006 | 0.017 | 0.011 |
| $D$ LEARNING RATE | 0.012 | 0.0004 | 0.021 | 0.082 | 0.012 | 0.019 | 0.006 |
| $G$ $\beta_1$ (ADAM) | 0.491 | 0.363 | 0.225 | 0.603 | 0.278 | 0.252 | 0.202 |
| $G$ $\beta_2$ (ADAM) | 0.319 | 0.752 | 0.331 | 0.614 | 0.777 | 0.931 | 0.975 |
| $D$ $\beta_1$ (ADAM) | 0.542 | 0.584 | 0.362 | 0.412 | 0.018 | 0.105 | 0.154 |
| $D$ $\beta2$ (ADAM) | 0.264 | 0.453 | 0.551 | 0.110 | 0.908 | 0.869 | 0.797 |
| EXPONENTIAL LR DECAY ($\gamma$) | 0.978 | 0.980 | 0.999 | 0.992 | 0.9996 | 0.985 | 0.996 |
| DECAY STEP SIZE | 3 | 19 | 15 | 16 | 11 | 13 | 17 |

Table 5: Hyperparameter Settings for DEQGAN (PDEs)

| HYPERPARAMETER | POS | HEA | WAV | BUR | ACA |
|---|---|---|---|---|---|
| NUM. ITERATIONS | 3000 | 2000 | 5000 | 3000 | 10000 |
| NUM. GRID POINTS | $32 \times 32$ | $32 \times 32$ | $32 \times 32$ | $64 \times 64$ | $64 \times 64$ |
| $G$ UNITS/LAYER | 50 | 40 | 50 | 50 | 50 |
| $G$ NUM. LAYERS | 4 | 4 | 4 | 3 | 2 |
| $D$ UNITS/LAYER | 30 | 30 | 50 | 20 | 30 |
| $D$ NUM. LAYERS | 2 | 2 | 2 | 5 | 2 |
| ACTIVATIONS | tanh | tanh | tanh | tanh | tanh |
| $G$ LEARNING RATE | 0.019 | 0.010 | 0.012 | 0.012 | 0.020 |
| $D$ LEARNING RATE | 0.021 | 0.001 | 0.088 | 0.005 | 0.013 |
| $G$ $\beta_1$ (ADAM) | 0.139 | 0.230 | 0.295 | 0.185 | 0.436 |
| $G$ $\beta_2$ (ADAM) | 0.369 | 0.657 | 0.358 | 0.594 | 0.910 |
| $D$ $\beta_1$ (ADAM) | 0.745 | 0.120 | 0.575 | 0.093 | 0.484 |
| $D$ $\beta2$ (ADAM) | 0.759 | 0.251 | 0.133 | 0.184 | 0.297 |
| EXPONENTIAL LR DECAY ($\gamma$) | 0.957 | 0.950 | 0.953 | 0.954 | 0.983 |
| DECAY STEP SIZE | 3 | 10 | 18 | 20 | 15 |

## A.6 Non-GAN Hyperparameter Tuning

Table 6 presents the minimum mean squared errors obtained after tuning hyperparameters for the alternative unsupervised neural network methods that use $L_1$, $L_2$ and Huber loss functions.

Table 6: Experimental Results With Non-GAN Hyperparameter Tuning

| Key | Mean Squared Error | | | | |
|---|---|---|---|---|---|
| | $L_1$ | $L_2$ | Huber | DEQGAN | Traditional |
| EXP | $1 \cdot 10^{-4}$ | $4 \cdot 10^{-8}$ | $2 \cdot 10^{-8}$ | $3 \cdot 10^{-16}$ | $2 \cdot 10^{-14}$ (RK4) |
| SHO | $1 \cdot 10^{-5}$ | $1 \cdot 10^{-9}$ | $5 \cdot 10^{-10}$ | $4 \cdot 10^{-13}$ | $1 \cdot 10^{-11}$ (RK4) |
| NLO | $1 \cdot 10^{-4}$ | $3 \cdot 10^{-10}$ | $1 \cdot 10^{-10}$ | $1 \cdot 10^{-12}$ | $4 \cdot 10^{-11}$ (RK4) |
| COO | $5 \cdot 10^{-1}$ | $2 \cdot 10^{-7}$ | $3 \cdot 10^{-7}$ | $1 \cdot 10^{-8}$ | $2 \cdot 10^{-9}$ (RK4) |
| SIR | $9 \cdot 10^{-6}$ | $1 \cdot 10^{-10}$ | $1 \cdot 10^{-10}$ | $1 \cdot 10^{-10}$ | $5 \cdot 10^{-13}$ (RK4) |
| HAM | $4 \cdot 10^{-5}$ | $1 \cdot 10^{-8}$ | $6 \cdot 10^{-9}$ | $1 \cdot 10^{-10}$ | $7 \cdot 10^{-14}$ (RK4) |
| EIN | $5 \cdot 10^{-2}$ | $2 \cdot 10^{-2}$ | $1 \cdot 10^{-2}$ | $4 \cdot 10^{-4}$ | $4 \cdot 10^{-7}$ (RK4) |
| POS | $9 \cdot 10^{-6}$ | $1 \cdot 10^{-10}$ | $1 \cdot 10^{-10}$ | $4 \cdot 10^{-13}$ | $3 \cdot 10^{-10}$ (FD) |
| HEA | $1 \cdot 10^{-4}$ | $4 \cdot 10^{-8}$ | $2 \cdot 10^{-8}$ | $6 \cdot 10^{-10}$ | $4 \cdot 10^{-7}$ (FD) |
| WAV | $4 \cdot 10^{-4}$ | $6 \cdot 10^{-7}$ | $2 \cdot 10^{-7}$ | $1 \cdot 10^{-8}$ | $7 \cdot 10^{-5}$ (FD) |
| BUR | $1 \cdot 10^{-3}$ | $1 \cdot 10^{-4}$ | $9 \cdot 10^{-5}$ | $4 \cdot 10^{-6}$ | $1 \cdot 10^{-3}$ (FD) |
| ACA | $5 \cdot 10^{-2}$ | $1 \cdot 10^{-2}$ | $3 \cdot 10^{-3}$ | $5 \cdot 10^{-3}$ | $2 \cdot 10^{-4}$ (FD) |

 **A.7   Residual Monitoring**

Figure 19 shows several examples of how we detect bad training runs by monitoring the variance of the $L_1$ norm of the $LHS$ (vector of equation residuals) in the first 25% of training iterations. Because the $LHS$ may oscillate initially even for successful runs, we use a patience window in the first 15% of iterations. In all three equations below, we terminate runs if the variance of the residual $L_1$ norm over 20 iterations exceeds 0.01.

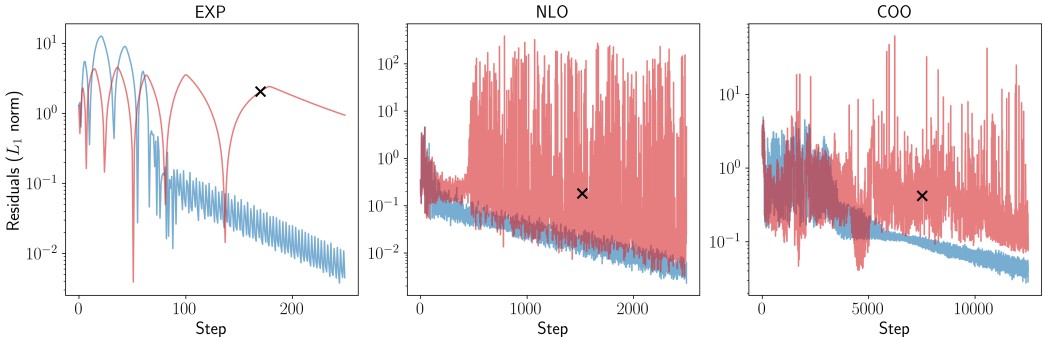

Figure 19: Equation residuals in the first 25% of training runs that ended with high (red) and low (blue) mean squared error for the exponential decay (EXP), non-linear oscillator (NLO) and coupled oscillators (COO) problems. The black crosses show the point at which the high MSE runs were terminated early.

