# OpenReview forum: "DEQGAN: Learning the Loss Function for PINNs with Generative Adversarial Networks"
_NeurIPS.cc/2022/Conference — NeurIPS 2022 Submitted_

### Official Review · Reviewer_rL1x · 2022-07-10

**Rating:** 7
**Confidence:** 3
**Soundness:** 3 good
**Presentation:** 4 excellent
**Contribution:** 3 good

**Summary:**

The paper proposes a method to solve ordinary and partial differential equations using GANs. Following in the tradition of physics-informed neural networks (PINNs), the flow equation is parametrized by a neural network. The parameters of the neural network are optimised by replacing a hand-tuned loss function such as the mean-squared error, with a discriminator and an adversarial loss. This method appears to provide improved performance over PINNs trained with hand-tuned loss functions and numerical solvers, on several well-known differential equations.

**Questions:**

As outlined in the previous section, I believe the paper makes a strong case for using GANs to solve differential equations. Some concrete suggestions to address the weaknesses:
1) It would be good to move the details of sections 3.2.1, 3.2.2 and 3.4 to the appendix, and move details of the ablation study to the main paper. Furthermore, sections 3.3 and 4.1 describe training / engineering choices to a level of detail that hinders readability, and can certainly be condensed.
2) It would be good to have a more extensive discussion of Figure 3. Moreover, Figure 3 appears to show only simulations from the GAN -- it would be good to have some examples in of the simulations from other methods as well, in order to contextualize the GAN performance beyond the reported MSE numbers.
3) While the methods residual monitoring and instance noise are certainly important details for reproducibility, it is not convincing that they would work without the other hyperparameter or architectural choices. It would therefore be good to modify the discussion of these two "methods", and to present them as potential avenues for stable training rather than a contribution of the paper.

Questions:
1) The performance of the different methods for solving differential equations are compared solely on the basis of MSE. Is there a particular reason for choosing this metric? -- it is particularly surprising that there is no justification provided for this in the paper, since the introduction makes a strong point that the choice of similarity metric typically used to optimize parameters for the PINNs are arbitrary. It would also be good to see whether the GAN performance is equally good under other similarity metrics e.g. L1, or MMD
2) In line 181-182, "We use the ReLU function...noise should not be used." -- this logic is not clear. Why should $L_g < L_d$ on any particular iteration indicate a better generator? -- this could also be due to initialisation, especially at the beginning of training.
3) How does the coarseness of the grid from which $t, x$ are sampled affect the performance of the GANs over other methods?

Minor comments:
1) The notation in section 4 is confusing in places: $G(x)$ and $\Psi_{\theta}(x)$ are used somewhat interchangeably; in line 179-181 $L_g$ and $L_d$ are not defined.
2) In Figure 2, a legend is not required in every panel
3) In Figure 3, the dotted lines in (a) and (b) are note identified in the legend; the colourbars have no label in (c) and (d)

**Limitations:**

It would be good if the discussion could address the following issues:
1) While the discriminator is not interpretable, the generator has potentially the same issues. What scientific insight is gained from an (effectively) black box generator modeling a flow equation?
2) While the GANs do provide an improvement over existing methods in terms of performance, it would be good to have an idea of the associated computational costs, sample efficiency and training times compared to the other methods.
3) How does this method scale with the number of dynamical variables, the size of the coarseness of the $t, x$ grid?

**Strengths And Weaknesses:**

Strengths:
1) The paper is very well-written, well-structured and easy to follow. The method and results, for the most part, are described in clear detail.
2) The ablation study, and discussion of methods to ensure stable training were extremely interesting and relevant, given that GANs are known to be difficult to train.
2) The extensive results in the paper make a convincing case for using GANs over other methods for solving differential equations -- in particular, with respect to the variability of the solutions from other methods across different problems.

Weaknesses:
1) Many of the details of training, hyperparameters etc., included in the methods section (sections 3.2.1, 3.2.2, 3.4), while important for reproducibility, distract from the main message of the paper, and hinder readability.
2) Some of the results require a lot more elucidation: e.g. the details of the ablation study in section 5.2 are mostly relegated to the appendix, and thus make the import of the results hard to parse; the plots in Figure 3 are barely mentioned or described in the text.
3) The paper could improve with a more extensive discussion of the limitations of the method (the conclusion mentions non-interpretability of the discriminator, but not much more).
4) The paper claims to present two "methods" to improve robustness in training: residual monitoring and addition of instance noise. While it is important to document the techniques used to stabilised training, neither of these are nescessarily novel contributions in the context of GANs. Furthermore, it is not clear whether these measures would be equally effective without spectral normalisation, skip connections in the generator, or with different choice of variance for the instance noise, or a change in the 25% training iteration threshold for residual monitoring.

---

> ### Author Response · Authors · 2022-08-02
> **Response to Reviewer rL1x**
>
> We appreciate the concrete suggestions and questions in this review. We would be happy to implement the modifications suggested to further improve the readability of the paper. To address the three questions posed by the reviewer:
>
> 1. We used MSE to evaluate the accuracy of PINN solutions in comparison to “ground-truth” solutions simply because this is a common and well-known metric, which we also believe is less arbitrary than using L2, L1, or some other loss that may be used in PINN optimization. We also used MSE to evaluate the accuracy of traditional numerical methods, enabling a fair comparison across all methods tested.
> 2. In our original implementation, we found that DEQGAN training sometimes failed when the discriminator loss plateaued below the generator loss, indicating that the generator was unable to fool the discriminator and neither model improved further. This motivated the addition of Gaussian noise to the “real” and “fake” data samples, which made the discriminator’s job more difficult and encouraged convergence to Nash equilibrium. A generator with a lower loss than the discriminator corresponds to the opposite scenario, in which the generator is already able to fool the discriminator.
> 3. The main effect of increasing the number of training points in PINN (and DEQGAN) optimization (i.e., using a finer $t,x$ grid) is an increase in computational cost. While using too few points might hinder solution accuracy, we generally found that the models achieved good accuracy with reasonably small grids, e.g. 32x32 or 64x64 for PDEs. We were also able to reduce interpolation error by sampling training points from noisy grids (evenly-spaced grids perturbed by Gaussian noise), which we mention in section 4 of the paper. As our main objective was to compare the performance of DEQGAN to classical PINNs in terms of accuracy, we used standard grid sizes for all methods. However, future work could investigate the relationships that exist among the number of training points, computational cost, and accuracy.
>
> To address the additional questions listed in the limitations section of the review:
>
> 1. The discriminator and generator networks in our method are no less interpretable than classical PINNs. Classical PINNs can be thought of as consisting of only a generator network, which is also a black-box model that is trained using a classical loss function such as L2, L1, or Huber. The addition of a discriminator model, therefore, does not hinder interpretability but, rather, offers possible insights that could be explored in future work. Investigating exactly what the discriminator network learns might enable better understanding of why some loss functions appear to be more effective for optimizing the generator than others.
> 2. Training time and computational complexity: While this is an important consideration, our primary objective was to show that DEQGAN can obtain more accurate solutions than classical PINNs on a wide variety of ODEs and PDEs, which necessitated a lengthy table of results (Table 2, page 7). Further, we found that all methods had similar runtimes, and therefore did not feel the need to make a comparison in our paper.
> 3. Size and coarseness of the $t,x$ grid: Addressed in (3), above.

---

> > ### Comment · Reviewer_rL1x · 2022-08-07
> > **Thank you**
> >
> > I thank the authors for their detailed response -- the answers clarified many things for me, and I am satisified that they answer most of my queries.
> >
> > Regarding point (2), and why one should expect the discriminator loss to be greater than the generator loss when the generator cannot fool the discriminator, I am still confused. Is my understanding correct, that this was an _empirical_ finding, and is not based on some theoretical reasoning? -- if yes, then I would suggest re-writing that last part of section 4.1 to reflect this. If not, it would really help to know how $L_g$ and $L_d$ are defined, and maybe a clearer explanation of the theoretical intuition behind this assertion.

---

> > > ### Author Response · Authors · 2022-08-08
> > > **Revised manuscript**
> > >
> > > Thank you for the follow up – we have posted a revised version of the paper that incorporates many of your helpful suggestions, including those on the overall structure of the paper. Regarding your question, we simply mean that a lower generator loss than discriminator loss indicates that the discriminator is generally performing worse than the generator, and we have softened the language in section 4.1 to reflect this. Thanks!

---

### Official Review · Reviewer_mpYu · 2022-07-11

**Rating:** 7
**Confidence:** 4
**Soundness:** 3 good
**Presentation:** 3 good
**Contribution:** 3 good

**Summary:**

The paper presents a new way to solve differentiable equations by leveraging GAN-based adversarial training to “learn” the loss function for PINNs.

**Questions:**

-It would be good to see an ablation study where they see the effectiveness of the “tricks” that they used to train the GAN effectively because some of them may not be needed.
-Can you make some comments on mode collapsing as a I mentioned on the weaknesses above?

**Limitations:**

The authors have addressed the limitations on the Conclusions section. The work is not applicable for negative societal impact in my opinion.

**Strengths And Weaknesses:**

Strengths
-Provides a novel method to solve Differential Equations with GANs in an unsupervised training setup. They do not use the solutions to the equations.
-As can be seen from the results it improves the performance on the Differential Equations task by multiple orders of magnitudes.
-They authors also integrate different effective methods in Section 3 for stable training of the GAN.
-Good paper presentation and a lot of details for the experiments in the Appendix. I also like the


Weaknesses
-No discussion for mode collapsing. I understand that this is a different application of GANs but it would be good to have a discussion on mode collapsing. Maybe the proposed GAN can provide different solutions for some equations.
-No theoretical analysis or guarantees are provided in the work. GANs in general they do not have a lot of guarantees but a theoretical analysis would be much appreciated.

---

> ### Author Response · Authors · 2022-08-02
> **Response to Reviewer mpYu**
>
> We thank the reviewer for these helpful comments and questions. While we did not use the term “mode collapse” in our paper, this issue and others related to GAN training instability were addressed in sections 4.1 and 4.2, which discuss the techniques we employed to improve the robustness of our method. We found that adding instance noise proportional to the difference between the generator and discriminator losses made convergence to equilibrium more likely, and we performed an ablation study to demonstrate this.

---

### Official Review · Reviewer_v1ox · 2022-07-11

**Rating:** 3
**Confidence:** 5
**Soundness:** 3 good
**Presentation:** 3 good
**Contribution:** 2 fair

**Summary:**

This work proposes Differential Equation GAN (DEQGAN) for solving differential equations using generative adversarial networks (GAN) where the generator learns to output the solution and the discriminator tries to learn a good loss function. The authors compare DEQGAN against classical PINNs over twelve ordinary differential equations (ODEs) and partial differential equations (PDEs). The experimental results show that DEQGAN can achieve multiple orders of magnitude lower L2 error than PINNs.

**Questions:**

1. The authors motivate the need of learning loss function by saying that L2,L1 loss functions lack theoretical justification. However, the author challenges the standard practice in physics-informed learning without explaining why and when L1, L2 loss functions can be bad choices. If L2 loss is problematic, how is the learned loss function better than L2 loss? Can the authors provide theoretical justification for it?
2. How fast is DEQGAN compared to PINNs and numerical solvers? Can authors provide time complexity comparison against PINNs and its improved variants such as SA-PINNs?
3. Only one specific artificial initial condition is used for each equation. How will DEQGAN perform if the initial condition is a random function sampled from a random field?

**Limitations:**

1. Lack of theoretical justification of the loss function and how the disciminator can learn a better objective function.
2. DEQGAN has training instability issue. To solve one equation, it needs to repeat runs multiple times and filters out the ones with poor performances.
3. The equations considered in the experiments are artificial. Only one specific initial condition is used for each equation, which raise the concern that the authors may pick the initial conditions to put DEQGAN in advantage in the experimental results.

**Strengths And Weaknesses:**

Strengths: the paper is well-written and easy to follow. DEQGAN demonstrates superior empirical results compared to classical PINNs. The fact that DEQGAN doesn't depend on predefined distance such as L1, L2 can be useful in some problems.

Weakness:
1. The Related Work section misses some important discussion [1, 2]. The authors should discuss the relationship to works and place the contribution of DEQGAN in the context of prior works. For example, SA-PINNs [1] can also be viewed as learning a loss function.
2. The authors state that the L2 loss lacks theoretical justification but miss the important explanation on why and when L2 loss can be problematic. Furthermore, DEQGAN do not have a theoretical justification of the learned loss function either.
3. The loss function in DEQGAN is borrowed from generative adversarial network (GAN). The original GAN work designs its loss function such that generator is minimizing Jensen-Shannon divergence between the generated distribution and target distribution given optimal discriminator dynamics. However, the Jensen-Shannon divergence doesn't make sense in DEQGAN because the problem is to solve a differential equation with unique solution. Even with Gaussian instance noise, minimizing the divergence between two Gaussians will just be minimizing the L2 loss. I do not see why a generative model is needed here.
4. The equations considered in the experiments are artificial. Only one specific initial condition is used for each equation, which raise the concern that the authors may pick the initial conditions to put DEQGAN in advantage in the experimental results.
5. DEQGAN has training instability issue. To solve one equation, it needs to repeat runs multiple times and filters out the ones with poor performances.

[1] McClenny, Levi, and Ulisses Braga-Neto. "Self-adaptive physics-informed neural networks using a soft attention mechanism." arXiv preprint arXiv:2009.04544 (2020).
[2] Daw, Arka, M. Maruf, and Anuj Karpatne. "PID-GAN: A GAN Framework based on a Physics-informed Discriminator for Uncertainty Quantification with Physics." Proceedings of the 27th ACM SIGKDD Conference on Knowledge Discovery & Data Mining. 2021.

---

> ### Author Response · Authors · 2022-08-02
> **Response to Reviewer v1ox**
>
> We appreciate the time taken by the reviewer to provide these detailed comments and thoughtful questions. However, we believe that this review does not fully appreciate the novelty of our method and contains several inaccuracies that we would like to address:
>
> 1. The reviewer speculates that specific initial conditions were chosen to highlight the advantage of DEQGAN – this is not the case. We used standard initial conditions for the differential equations considered and emphasize that DEQGAN performs similarly across other values. We did not include results for multiple initial conditions because our aim was to showcase results on a variety of equations that exhibit challenging and varied dynamics.
> 2. The reviewer suggests that multiple training runs are required to solve a single differential equation – this is also not the case. While our original formulation of DEQGAN required hyperparameter tuning to attain the best results, we proposed two methods to improve robustness (instance noise and residual monitoring) and conducted an ablation study to demonstrate the efficacy of these methods. In this study, we performed 500 training runs on a single equation and showed that for the vast majority of hyperparameter values, DEQGAN performs very well. In practice, however, only a single training run is required to obtain an accurate solution.
> 3. The reviewer notes that we do not provide a theoretical explanation of when or why classical loss functions like L1 and L2 perform poorly. These are good questions, but they are out of scope for this work and remain open research problems. Our method circumvents this gap in the theory by proposing an adversarial training setup that can be thought of as learning the loss function for the generator, but we do not discredit any work that aims to address this question more directly. In fact, our conclusion suggests that future work could examine exactly what is learned by the discriminator network of DEQGAN, which might help us understand why some loss functions appear to be more effective for optimizing the generator than others. We also note that although a theoretical explanation is currently lacking, our empirical results clearly demonstrate that classical loss functions (L2, L1, and Huber) show varied performance and that DEQGAN consistently obtains more accurate solutions.
> 4. The reviewer cites self-attention PINNs as an alternative method that “learns a loss function.” While the loss proposed in SA-PINNs incorporates self-adaptation weights to improve training, the losses over these terms still take the form of L2 (equations 11-13 in their paper), making this method similar to other adaptive losses, which we also cite in our paper [1]. We would be happy to mention SA-PINNs in our related works section for completeness.
> 5. The reviewer asks about the time complexity of DEQGAN in comparison to classical PINNs. While this is an important consideration, our primary objective was to show that DEQGAN can obtain more accurate solutions than classical PINNs on a wide variety of ODEs and PDEs, which necessitated a lengthy table of results (Table 2, page 7). Further, we found that all methods had similar runtimes, and therefore did not feel the need to make a comparison in our paper.
>
> [1] Zeng, S., Zhang, Z., & Zou, Q. (2022). Adaptive deep neural networks methods for high-dimensional partial differential equations. Journal of Computational Physics, (pp. 111232).

---

> > ### Comment · Reviewer_v1ox · 2022-08-06
> > **Some clarifications**
> >
> > Thank authors for their quick response. I would like to clarify several points.
> > 1. The definition of 'the standard initial condition' mentioned in authors' response is not clear. To me, one specific initial condition for each equation cannot provide strong evidence for an empirical paper. I would still suggest do experiments on multiple initial conditions sampled from a random field because the experiments should be extensive to better understand the proposed method and how they will perform under different conditions.
> > 2. I also would like to elaborate more on the second point in the weakness section. We know for a big class of problems L2 loss is reasonable with theoretical justifications [3]. My point is that authors should be careful when they say standard L2 loss lack of theoretical justification and should be more specific about the problem settings. In fact, many PDEs (eg. heat equation, viscous Burgers equation) in DEQGAN's experiments belong to this class of problems where L2 has theoretical justification. In contrast, DEQGAN does not have theoretical justification even for those problems. It is mysterious to me how generative model fits in here and how the loss objective is chosen in the paper. It seems to be copying the GAN framework to PINN training without any theoretical justification for why generative model makes sense here.
> > 3. By SA-PINNs I mean the self-adaptive physics-informed neural networks [4]. The paper cited by authors is some other paper which is also related but not what I meant. Self-adaptive PINNs perform much better than classical PINNs and their loss is adaptively learned. That's why I think it is an important baseline.
> > 4. Regarding the second point in the response, I appreciate author's clarification and ablation study in Table 3. I agree that the ablation study Table 3 shows DEQGAN with residual monitor and instance noise can resolve instability issue.
> >
> > [3] Mishra, Siddhartha, and Roberto Molinaro. "Estimates on the generalization error of physics-informed neural networks for approximating a class of inverse problems for PDEs." _IMA Journal of Numerical Analysis_ 42.2 (2022): 981-1022.
> >
> > [4] McClenny, Levi, and Ulisses Braga-Neto. "Self-adaptive physics-informed neural networks using a soft attention mechanism." _arXiv preprint arXiv:2009.04544_ (2020).

---

> > > ### Author Response · Authors · 2022-08-08
> > > **Follow up on clarifications**
> > >
> > > Many thanks for the follow up comment. We have posted a revised version of the paper that incorporates many of your helpful comments and would like to provide additional clarifications:
> > >
> > > 1. By “standard initial conditions,” we simply mean values that are commonly used in the context of particular differential equations. For example, for the damped nonlinear oscillator (NLO) problem, we use $x_0=1, \dot{x}_0=0.5$ simply because these are clean values. However, they are also arbitrary in the sense that we could have trained DEQGAN on different initial conditions and obtained similar results. In Figure 4a of the updated paper (page 9), we have plotted the phase space of the DEQGAN solutions (solid color lines) for three different initial velocities $\dot{x}_0=0.5, 0.6, 0.7$ and can see that these are indistinguishable from the solutions obtained using a high-quality numerical integrator (dashed black lines). We emphasize that DEQGAN attains highly accurate solutions on arbitrary initial conditions, but we believe that it is much more compelling to showcase its performance on a variety of equations (we include twelve) that exhibit different and challenging dynamics.
> > > 2. Thank you for elaborating on the point about theoretical justification. As our work is focused on the predictive performance of various PINN methods, we mean that no particular loss function has been demonstrated to provide advantages in terms of convergence or solution accuracy. However, our empirical results indicate that classical loss functions (L2, L1, and Huber) show varied performance on different problems and that DEQGAN consistently achieves better predictive accuracy. The paper [3] cited by the reviewer focuses on generalization error, and while this may provide a reason (e.g. uncertainty quantification) to use a particular loss function, it is not the reason we are interested in. We agree that we could have made this point more clear and have updated the third paragraph of the introduction (page 1) to reflect this.
> > > 3. The paper on SA-PINNs cited by the reviewer [4] is the same one we were referring to in our original comment – the additional citation we provided was to a similar paper [5] on adaptive loss functions that we had referenced in our paper. While the losses proposed in [4, 5] are adaptive, we still view them as fundamentally different from DEQGAN because our method uses a neural network (the discriminator) to learn the loss function for optimizing the generator and is therefore much more flexible than an explicit loss function. Nonetheless, we agree that these works should be referenced in our paper and have updated the fourth paragraph of the introduction to include them. We believe that our empirical results provide strong evidence that it would be worthwhile for future work to investigate exactly what the discriminator network learns, which might elucidate why some loss functions achieve better predictive performance than others.
> > >
> > > [3] Mishra, Siddhartha, and Roberto Molinaro. "Estimates on the generalization error of physics-informed neural networks for approximating a class of inverse problems for PDEs." IMA Journal of Numerical Analysis 42.2 (2022): 981-1022.
> > >
> > > [4] McClenny, Levi, and Ulisses Braga-Neto. "Self-adaptive physics-informed neural networks using a soft attention mechanism." arXiv preprint arXiv:2009.04544 (2020).
> > >
> > > [5] Zeng, S., Zhang, Z., & Zou, Q. (2022). Adaptive deep neural networks methods for high-dimensional partial differential equations. Journal of Computational Physics, (pp. 111232).

---

### Comment · Area_Chair_WCAZ · 2022-08-09
**Concern on the paper.**

Hi authors,

I read the paper and have three concerns:

(1) The literature survey in this paper is insufficient.  The idea of using GAN-type methods to solve PDEs is not new. For example, there have been some work in the applied mathematics literature: https://arxiv.org/pdf/1907.08272.pdf and https://arxiv.org/pdf/2002.11309.pdf, which are not cited in this paper.

(2) The paper only compared their method with basic PINN and very low dimensional PDEs. They should at least compare with https://arxiv.org/pdf/1907.08272.pdf and higher dimensional PDEs.

(3) The authors argue that existing methods do not have theoretical justifications on the loss functions. However, I did not find any theoretical justification in their paper also.

Could you please make some clarification?

Best

AC

---

> ### Author Response · Authors · 2022-08-09
> **Follow up to AC**
>
> Many thanks to the AC for the comments, and we'd be happy to clarify:
>
> 1. We are aware that we are not the first to apply GANs to solving differential equations and cite multiple works that do this in the Related Work section (third paragraph). The first paper cited by the AC [1] leverages the weak form of PDEs by training the generator and discriminator to approximate the weak solution $u_{\theta}$ and test function $\phi_{\eta}$, respectively. By contrast, our method (DEQGAN) is based on the strong form, which frees the discriminator to learn the loss function for optimizing the generator. We also note that this paper addresses only a small number of PDEs and does not always outperform classical PINNs, whereas DEQGAN consistently outperforms classical PINNs on a suite of twelve problems (including PDEs and ODEs). The second paper cited by the AC [2] is narrowly focused on a particular linear equation and does not directly leverage GANs.
> 2. While our results do not include high-dimensional PDEs, we show that DEQGAN consistently outperforms classical PINNs on a wide variety of challenging equations, many of which exhibit higher degrees of non-linearity and more complex dynamics than those addressed in [1]. Given our promising results, we expect our method to also perform well on higher dimensional problems and think that this would be a worthwhile direction for future work.
> 3. Our approach replaces an explicit loss function (e.g., L2, L1, Huber) with a discriminator network that learns the loss function for optimizing the generator. Therefore, DEQGAN circumvents the lack of theoretical justification for using any particular loss function entirely and offers the flexibility to overcome the weaknesses of explicit loss functions. Indeed, our results make clear that these losses show variable performance on different equations (sometimes failing entirely) and are consistently outperformed by DEQGAN in terms of predictive accuracy. These results indicate that it would be worthwhile for future work to examine exactly what is learned by the discriminator network of DEQGAN, which might help us understand why some loss functions appear to be more effective for training PINNs than others.
>
> [1] Zang, Yaohua, et al. "Weak adversarial networks for high-dimensional partial differential equations." Journal of Computational Physics 411 (2020): 109409.
>
> [2] Liu, Shu, et al. "Neural Parametric Fokker--Planck Equation." SIAM Journal on Numerical Analysis 60.3 (2022): 1385-1449.

---

### Comment · Area_Chair_WCAZ · 2022-08-10
**Rebuttal Acknowledgement**



Dear Reviewers,

We are entering the discussion phase, where the authors will be not involved in the discussion.

I would like to request you to confirm that you have already read the rebuttal from the authors.

Best

AC

---

### Meta-Review · Area_Chair_WCAZ · 2022-08-26

**Recommendation:** Reject
**Confidence:** Certain

**Metareview:**

This paper presents a new method for solving differential equations using generative adversarial networks to "learn the loss function" for optimizing the neural network. After the discussion, the reviewers still have a few major concerns:

(1) The authors' claim that the existing methods lack of theoretical justifications. However, the paper does not provide a sufficient justification on their proposed method, either, which makes their key motivation of the paper questionable.

(2) Some important baseline methods are missing in the comparison as well as references. The authors should improve their literature survey.

(3) The computational challenges of solving PDEs mainly lie in high dimensionality. Most of existing deep-learning based PDE solves, including PINN, attempt to demonstrate the benefit of using deep neural networks for approximating high dimensional functions or operators. However, the experiments only consider low dimensional PDEs, which not difficult to solve, and existing numerical methods can solve them efficiently without deep neural networks and complicated tuning.

**Award:**

No

---

### Decision · Program_Chairs · 2022-09-14

Reject